

# Sensitivity of precipitation in the highlands and lowlands of Peru to physics parameterization options in WRFV3.8.1

Santos J. González-Rojí[1,2], Martina Messmer[1,2], Christoph C. Raible[1,2], and Thomas F. Stocker[1,2]

[1]Climate and Environmental Physics, University of Bern, Bern, Switzerland
[2]Oeschger Centre for Climate Change Research, University of Bern, Bern, Switzerland

**Correspondence:** Santos J. González-Rojí (santos.gonzalez@unibe.ch)

**Abstract.** The performance of the Weather Research and Forecasting (WRF) model version 3.8.1 at convection-permitting scale is evaluated by means of several sensitivity simulations over southern Peru down to a grid resolution of 1 km, whereby the main focus is on the domain with 5 km horizontal resolution. Different configurations of microphysics, cumulus, longwave radiation and planetary boundary layer schemes are tested. For the year 2008, the simulated precipitation amounts and patterns
are compared to gridded observational data sets and weather station data gathered from Peru, Bolivia and Brazil. The temporal correlation of simulated monthly precipitation sums against in-situ and gridded observational data show that the most challenging regions for WRF are the slopes along both sides of the Andes, i.e., elevations between 1000 and 3000 m above sea level. The pattern correlation analysis between simulated precipitation and station data suggests that all tested WRF setups perform rather poorly along the northeastern slopes of the Andes during the entire year. In the southwestern region of the domain the
performance of all setups is better except for the driest period (May–September). The results of the pattern correlation to the gridded observational data sets show that all setups perform reasonably well except along both slopes during the dry season. The precipitation patterns reveal that the typical setup used over Europe is too dry throughout the entire year, and that the experiment with the combination of the single-moment 6-class microphysics scheme and the Grell–Freitas cumulus parameterization in the domains with resolutions larger than 5 km, suitable for East Africa, does not perfectly apply to other equatorial
regions such as the Amazon basin in southeastern Peru. The experiment with the Stony–Brook University microphysics scheme and the Grell-Freitas cumulus parameterization tends to overestimate precipitation over the northeastern slopes of the Andes, but allows to enforce a positive feedback between the soil moisture, air temperature, relative humidity, mid-level cloud cover and finally, also precipitation. Hence, this setup is the one providing the most accurate results over the Peruvian Amazon, and particularly over the department of Madre de Dios, which is a region of interest because it is considered the biodiversity hotspot
of Peru. The robustness of this particular parameterization option is backed up by similar results obtained during wet climate conditions observed in 2012.



## 1 Introduction

Tropical regions are known for their high level of biodiversity (Lamoreux et al., 2006), but also for their rather complex weather
and climate conditions. In the case of Peru, this complexity is caused by the surrounding oceans, lakes, topography and diverse
vegetation. Additionally, the country is subject to several different climate zones, such as the arid Pacific coastal deserts, the
temperate very steep slopes of the Andean mountain ranges, and the low-lying tropical rainforest in the Amazon basin (Beck
et al., 2018). Precipitation over the Amazon is mainly driven by the South American monsoon, which takes place during austral
spring and summer (Marengo et al., 2012; Cai et al., 2020). The monsoon starts after the establishment of a pressure dipole: the
Bolivian high pressure system, which is induced by diabatic heating over the Amazon (Lenters and Cook, 1997), and the low
pressure system over the Chaco region (over Bolivia, Paraguay and northern Argentina), which is initiated by surface heating
(Vera et al., 2006). In combination with the subtropical high over the southern Atlantic Ocean, these three pressure systems
intensify the trade winds, boosting the moisture flow towards the Amazon basin, which results in convergence and therefore
in precipitation (Carvalho et al., 2004; Marengo et al., 2012). Blocked by the Andes, the flow is channeled towards the south
and establishes the South American low-level jet (Marengo et al., 2004). This low-level jet can trigger convective systems
over southern South America (Salio et al., 2007). Over the western tropical coasts of Ecuador and Peru, the rain is mainly
forced by El Niño–Southern Oscillation (ENSO), as the heat from the warm ocean waters is transferred to the atmosphere
inducing convection and heavy precipitation. This is usually observed in February and March, when the maximum sea surface
temperature anomalies are observed over the Pacific Ocean (Hu et al., 2019). The effect of ENSO is not only restricted to
the western coasts of tropical South America, and its influence is also perceptible over the entire continent. For example, the
atmospheric part of ENSO, i.e., the Walker circulation over the Pacific ocean, modifies the location of the main ascent and
descent regions (Grimm, 2003; Ropelewski and Bell, 2008; Sasaki et al., 2015), and thus alters the advection of moist and
warm air towards the continent (Rutllant and Fuenzalida, 1991).

As a consequence of the combined effect of these drivers, regional differences are observed in the seasonal precipitation
over Peru. The rainforest receives abundant precipitation during the austral summer (November to March) and little amounts
during the rest of the year (April to October). A similar rainy season is also observed along the mountain ranges. However,
near the coast, precipitation is rather scarce throughout the year, with the exception of the northwestern coast where rain falls
from December to March due to the direct influence of ENSO in the region (Rau et al., 2017; Sanabria Quispe, 2018). ENSO
also contributes to the variability of rainfall in the highlands (Imfeld et al., 2019). Recent droughts in northeastern Peru were
induced by anomalously warm Atlantic sea surface temperatures, which led to a displacement of the intertropical convergence
zone towards the north, causing anomalously dry conditions especially during the dry season in the austral winter (Espinoza
et al., 2011). This displacement is consistent with a negative trend in precipitation amounts and with a positive trend in the
number of dry days over the southern Amazon basin from September to November observed during the period 1981–2017
(Espinoza et al., 2019). In contrast, higher precipitation amounts and fewer dry days are recorded in the northern Amazon
basin from March to May for the same period (Espinoza et al., 2019). Hence, cold anomalies in sea surface temperatures
over the Atlantic Ocean are drivers of extreme precipitation and floodings over the southwestern Amazon basin, while cold



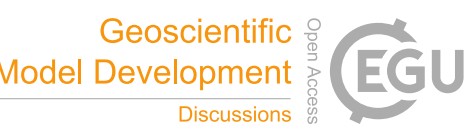

anomalies in the Pacific lead to extreme precipitation in the northeastern slopes of the Bolivian Andes (e.g. Ronchail et al., 2005; Molina-Carpio et al., 2017). These extreme events have strong impacts on society, ecosystems and economy of the region. The most common ones are damages to houses, infrastructure and croplands, fatalities (Rodríguez-Morata et al., 2019; Takahashi and Martínez, 2019), and even increases in disease transmission (Poveda et al., 2001).

One important source to characterize the hydrological cycle over Peru and to gain understanding of the related processes are observations. The number of weather stations in Peru depends on the region considered. While a dense network exists along the coast and the Peruvian Andes, the number of stations in the Amazon basin is rather limited. This is particularly the case for the Department of Madre de Dios in Peru, a region in the southeastern part of the country located in the low-lying rain-

forest, bordering Bolivia and Brazil. The lack of weather station data hampers the understanding of the atmospheric processes and their interactions which lead to the aforementioned complex climate. Furthermore, it also complicates the projection of changes related to global warming, and hence to socio-economic developments in the region. Climate models, and in particular regional climate models, can contribute to a better understanding of those processes. However, simulating weather and climate conditions over tropical regions is complicated and challenging for regional climate models, and hence, their performance must

be evaluated first (e.g. Rauscher et al., 2010; Brune et al., 2020).

The number of climate simulations over South America has been growing in recent years, and one of the most commonly used models is the state-of-the-art Weather Research and Forecasting (WRF) model (Skamarock et al., 2008). For example, Müller et al. (2016) performed a simulation over the entire continent at 45 km resolution with a nested domain at 15 km over the La Plata basin to test the predictability of the model over that region. The results show a skillful forecast of precipitation and

temperature up to seven days, in particular over humid and temperate climates. Another simulation at 25 km resolution over northern South America and the Caribbean Sea shows that positive biases in precipitation originate from a misrepresentation in the regional flow and Pacific sea surface temperatures (Martinez et al., 2019). In the case of Peru, most of the regional simulations focus on the Andes and the western slopes. Mourre et al. (2016) suggested that the topography and the precipitation gradients are so complex that simulations at spatial resolutions of 9 km are still not good enough to capture the precipitation

amounts and patterns over the Cordillera Blanca. This is in line with Moya-Álvarez et al. (2019), who studied the effect of the spatial resolution on the precipitation forecasts over the central Andes of Peru. They found an improvement in the simulation of accumulated precipitation over 10-day periods when increasing the resolution of the model from 18 to 3 km. However, the model is not able to correctly represent the internal structure of a local hailstorm and its rainfall forecast until reaching a spatial resolution of 750 m (Moya-Álvarez et al., 2019).

Thus, it seems clear that convection-permitting spatial scales are necessary to simulate correctly the precipitation patterns in such complex regions. Additionally, the dominant dynamics in the region are not straightforward, and consequently, the models must be tested and tuned to determine the best setup for each region. Several studies investigated the interchangeability of the parameterizations of the models from one region to another (e.g. Takle et al., 2007; Jacob et al., 2012; Russo et al., 2020), but there is still no clear answer for it, in particular in the case of convection-permitting scales. In this study, we focus on the region

of Madre de Dios, a part of Peru located where the steep complex topography meets the low-lying rainforest of the Amazon basin. This region is considered one of the richest regions in Peru in terms of biological and cultural diversity (Killeen, 2007),





and the economy of this region heavily depends on the natural resources and raw materials. However, mining and deforestation have been exacerbated since 2005 by the expansion of the Interoceanic Highway (Fisher et al., 2018; Sánchez-Cuervo et al., 2020). Albeit the region has a high economic value for Peru, weather station data and records of atmospheric circulations are scarce, and also regional modelling studies on the transition zone between the mountains and the rainforest are missing. It must be assumed that the current changes in land use and atmospheric dynamics in the region will be aggravated because of climate change (IPCC, 2013). Hence, it is timely to provide high-resolution regional climate model experiments that pave the way for a deeper understanding of the atmospheric circulation and interactions with the vegetation, and to be able to understand effects of deforestation and global warming in the "Biodiversity Capital of Peru" (Gobierno Regional de Madre de Dios, 2014).

The structure of the paper is the following: first, information about the sensitivity simulations performed with WRF is given in Sect. 2, along with the details of the weather station data and the observation-based gridded data sets used in the analysis. The results from the evaluation to determine the most accurate set of parameterizations over the focus region are presented and discussed in Sect. 3. Sect. 4 provides a summary and concluding remarks.

## 2 Model configuration and Data

### 2.1 WRF model

The WRF model (version 3.8.1) is used to generate a set of simulations over Peru. Initial and boundary conditions necessary to run the model are provided by the reanalysis data of the European Centre for Medium-Range Weather Forecast: ERA5 (Hersbach et al., 2020). Even though the data are available every hour at a spatial resolution of 0.25°, only 6 hourly data are fed into the model to allow it to develop with a certain level of freedom. 24 out of the 37 provided vertical pressure levels were considered for running the simulations (from 10 to 1000 hPa). ERA5 is suitable to provide initial and boundary conditions as recent studies showed realistic spatial patterns of the temperature and precipitation over South America improving the precursor product ERA-interim (Balmaceda-Huarte et al., 2021) and reduced biases compared to other reanalyses (Hassler and Lauer, 2021).

A triple-nested domain setting is used for the simulations with a 1:5 ratio (see Figure 1). This approach allows us to cover the northern part of South America with a spatial resolution of 25 km. The second domain zooms in towards the southern part of Peru, covering parts of the Pacific coast, the transition zone between the mountain ranges in the Andes, and the rainforest at a horizontal resolution of 5 km. Finally, the innermost domain focuses on the flatlands of Madre de Dios with a spatial resolution of 1 km.

The main analysis is based on the year 2008. The simulations are initialized on the 1[st] of November 2007, and end on the 31[st] of December 2008. The first two months are the spin-up period and hence, are not included in the analysis. The soil variables in ERA5 are assumed to be in quasi-equilibrium, so a spin-up period of two months is enough to balance the fluxes between the atmosphere and soil in WRF. This is backed up by previous studies showing that short spin-up periods are sufficient for precipitation analyses performed with WRF (e.g. Angevine et al., 2014; Jerez et al., 2020; Velasquez et al., 2020). Compared to the climatology of Madre de Dios for the year 1981–2010, 2008 is a normal year in terms of precipitation. Moreover, 2008 also





125 shows a well-differentiated rainy and dry season, and none of the months are extremely wet or dry compared to the climatology (Figure 2). El Niño 3.4 index shows that the beginning of 2008 is at the end of a strong La Niña event, while the middle of the year is in a more or less neutral state, which transforms back to a moderate La Niña state at the end of the year (Trenberth and National Center for Atmospheric Research Staff, 2020). Note that at least Madre de Dios is not very strongly affected by the phases of ENSO in terms of precipitation variability, which is particularly true for the rainy season at the beginning and end of

130 the year, where La Niña phases are observed (Cai et al., 2020; Lindsey, 2016).

  Several physics options included in WRF are tested to obtain a reliable representation of precipitation over the southern part of Peru. To do so, some of the microphysics, long-wave (LW) radiation and planetary boundary layer (PBL) schemes are used in separate experiments. The cumulus parameterization is also changed depending on the experiment, but it is switched off in the domains with spatial resolutions equal to or smaller than 5 km, i.e., in domains D2 and D3, as convective precipitation

135 can be explicitly resolved by the model at these resolutions. The WRF single-moment 6-class scheme (WSM6; Hong and Lim, 2006) and the Stony–Brook University scheme (SBU; Lin and Colle, 2011) are the two options tested for microphysics. The Rapid Radiation Transfer Model (RRTM; Mlawer et al., 1997) and Community Atmosphere Model (CAM; Collins et al., 2004) are considered for the LW radiation schemes. For the PBL schemes, the non-local diffusion scheme of Yonsei University (YSU; Hong et al., 2006) and the Asymmetric Convection Model 2 (ACM2; Pleim, 2007) are tested. The Kain-Fritsch (Kain,

140 2004) and the scale-aware Grell-Freitas ensemble (Grell and Freitas, 2014) schemes are selected for the parameterization of cumulus processes in the coarsest domain D1. The remaining parameterizations are kept constant between all the sensitivity experiments. These physics options include the Noah-MP land surface model (Niu et al., 2011; Yang et al., 2011), the Dudhia short-wave (SW) scheme (Dudhia, 1988) for SW radiation and the CLM4.5 lake model (Subin et al., 2012). The lake model is switched on, because it is expected to improve the simulation of the interaction between the lake surface and the atmosphere

145 (e.g., heat transfer), and consequently, to better represent evaporation and precipitation (Gu et al., 2015).

  A summary of the parameterizations used in each experiment is presented in Table 1. The naming of each experiment is based on the main parameterization employed or on the region where the setting is usually applied (first column, Table 1). Based on previous studies by the authors, the "Europe" experiment includes the updated parameterizations used over that region (Messmer et al., 2017), i.e., Noah-MP instead of Noah land surface scheme. Additionally, the "Kenya" experiment is the same as the

150 one which is most successful in simulating precipitation and temperature in the surroundings of Mount Kenya (Messmer et al., 2021). The comparison against this experiment will help us to determine if the setups over equatorial regions are interchangeable. The "South America" experiment takes as a reference the parameterizations used to simulate storms over the central Andes (Zamuriano et al., 2019). The "Micro13" and the "No Cumulus" experiments are based on the parameterizations of the Kenya experiment, but with a different microphysics scheme and without any cumulus parameterization, respectively. The

155 No Cumulus experiment is motivated by improvements obtained for precipitation when simulating convective precipitation explicitly at a spatial resolution of 25 km or below (Vergara-Temprado et al., 2020), and the good performance in simulating precipitation and temperature over Mount Kenya (Messmer et al., 2021).

  In addition to year 2008, the year 2012 was selected to test the performance of the model under wet climate conditions. The same strategy applied to year 2008 was followed for 2012, but only parameterization options showing a good performance





in 2008 were included. 2012 is a wet year and presents a well-differentiated rainy and dry season, compared to a 30-year climatology of Madre de Dios (Fig. 2). The El Niño 3.4 index shows that the first months of the year are influenced by the end of a weak La Niña event, while the rest of the year is in a more or less neutral state (Trenberth and National Center for Atmospheric Research Staff, 2020).

## 2.2 Observational data

The simulated precipitation is compared with weather station and gridded observational data sets. Weather station data from Peru, Bolivia, and Brazil are considered. The weather station data from Peru are provided by the Servicio Nacional de Meteorología e Hidrología (SENAMHI) del Perú, the data from Bolivia by the SENAMHI Bolivia, and the data from Brazil by the Instituto Nacional de Meteorologia (INMET). Some additional weather station data were retrieved from the World Weather Records (WWR) database from the World Meteorological Organization (WMO). The locations of the selected 175 weather

stations for the year 2008 are depicted in Fig. 1. The station data are separated into five regions according to two characteristics; their elevation and their location with respect to the plateau. The stations between 0 and 1000 m above sea level (a.s.l.) are considered to be in the flatlands (depicted as circles in Fig. 1); stations between 1000 and 3000 m a.s.l. are located along the slopes (stars); and weather stations higher than 3000 m a.s.l. represent the Andean plateau (triangles). The flatland and the slopes are further separated into southwestern (SW) and northeastern (NE) parts with respect to the plateau. 175 weather

stations are distributed over the entire domain in 2008, but the number of stations in each category varies: nine in the SW flatlands, 24 in the SW slopes, 120 in the plateau, seven in the NE slopes, and 15 in the NE flatlands. For the wetter year 2012, the number of stations is slightly reduced to 169: eight in the SW flatlands, 24 in the SW slopes, 119 in the plateau, seven in the NW slopes, and 11 in the NE flatlands. Further details about the stations are presented in the corresponding files submitted to Zenodo (see the code and data availability section).

Apart from the weather station data, the precipitation from the driving reanalysis ERA5 is also included in the analysis so that we can evaluate how good the performance of ERA5 is at simulating precipitation amounts and patterns over the region. As suggested by recent studies (e.g., Hassler and Lauer, 2021; Rivoire et al., 2021), satellite or weather station based gridded data sets should be considered as a reference in the validation process instead of reanalyses. Thus, several gridded observational data sets are employed in our study such as the Tropical Rainfall Measurement Mission (TRMM), the Integrated Multi-satellitE

Retrievals from GPM (IMERG), the Climate Hazards group Infrared Precipitation with Stations (CHIRPS) and the Peruvian Interpolated data of SENAMHI Climatological and hydrological Observations (PISCO). The separation into the five regions, discussed in the description of the weather station data above, is also applied to the gridded observational data sets, ERA5 reanalysis and the WRF simulations.

TRMM is a gridded product with a 0.25° spatial and 3-hourly temporal resolution for the period 1998–2019 and for the

area 50°N to 50°S. We use the TRMM 3B42 (version 7; Liu, 2015) as recommended by NASA and the Japanese Aerospace Exploration Agency (JAXA) for scientific research. This data set combines satellite data (3-hourly resolution) with rain gauge data from the Global Precipitation Climatology Centre (GPCC) for the monthly sums. As IMERG (Huffman et al., 2019) is the successor of TRMM, the data are produced in the same way, but the temporal and spatial resolution is finer. The IMERG





product provides half-hourly precipitation with a spatial resolution of 0.1° (approximately 10 km). Version 6 is employed in
this study, which is available for the period 2000–present.

CHIRPS is a high-resolution precipitation data set that covers the area 50°N to 50°S (Funk et al., 2015). The daily precipitation amounts are available at a 0.05° spatial resolution for the period 1981–present. As the previous data sets, it combines satellite data with the World Meteorological Organization's Global Telecommunications System (GTS) rain gauge data.

PISCO Version 2.1 provides land-only daily precipitation amounts estimated for the entire country of Peru at a spatial resolution of 0.1° for the period 1981–2018. This precipitation data set combines radar and comparably dense gauge measurements (441 quality controlled stations) maintained by SENAMHI with the CHIRPS data set. The performance of this product is evaluated against independent weather stations from those used to develop PISCO, and the coast and the western slopes of the Andes showed the best scores (Aybar et al., 2020). These regions coincide with areas covered by the highest weather station density.

## 3 Results

### 3.1 Temporal analysis of precipitation

To evaluate the performance of the different setups of the WRF model, we start by analyzing the annual cycle according to the monthly precipitation sums of year 2008. Spearman correlation coefficients (Fig. 3) and root-mean-square error (RMSE; Fig. 4) are depicted in box and whisker plots and maps. The indices depicted in the box and whisker plots are calculated for each weather station data and its closest respective grid point in the second domain of the WRF grid. For this, all the gridded observation based data are previously bi-linearly interpolated to the respective WRF grid using the function of the Climate Data Operator (CDO; Schulzweida, 2019). For the maps, all the gridded data are bi-linearly interpolated to the original grid of PISCO before calculating the two statistics between the gridded data (observation based and WRF) and the PISCO data set. PISCO is used as reference because it is generated particularly for Peru by combining weather station data with satellite data (Sect. 2.2). Box and whisker plots are estimated for different regions (Sect. 2.2): SW or NE flatlands, SW or NE slopes, and the plateau.

The temporal correlation coefficients in the box and whisker plots show that the area with the highest correlation is the plateau, where the median values of all the correlation coefficients are above 0.8 (Fig. 3a). PISCO shows the highest median value with the smallest spread. The good agreement between the weather station data and PISCO confirms the good quality of the latter in the plateau, related to the available dense station network. The high temporal correlations between the weather station data and the simulations with the different parameterization options are confirmed by the correlation maps against PISCO (Fig. 3f–l).

The box and whisker plots reveal also a high temporal correlation in the two slope regions (median $r > 0.75$). For the SW slopes (Fig. 3b) the distribution of the correlation coefficients is rather homogeneous, with a slightly lower median for Micro13, however, not significantly different to the others. This is reversed in the NE slopes (Fig. 3c), where Micro13 shows similar correlations to those for the gridded observational data sets, and Kenya, No Cumulus and in particular Europe present



slightly lower correlation coefficients. High temporal correlations between PISCO and all other gridded data (Fig. 3f–l) are also observed in the slopes at both flanks of the Andes, depicted in the maps. An exception is the region to the south in the SW slopes where the correlation coefficients with respect to PISCO are close to zero for all other gridded data sets. The fact that

CHIRPS, IMERG and the sensitivity simulations correlate better with the weather station data in this southern part implies that the PISCO data set is not fully able to realistically represent observed precipitation in this region.

The lowest temporal correlation coefficients in the box and whisker plots are found for the flatlands. In the case of the SW flatlands (Fig. 3d), the correlation coefficients between the weather station data and the WRF simulations are in the range of those obtained by the gridded observational data sets, whereby Kenya and No Cumulus show the worst performance. The

correlation coefficients of Micro13 with respect to the weather station data are comparable to the ones obtained by CHIRPS or IMERG. Europe and South America show the best performance in the SW flatlands, i.e., in an extremely dry area, because these two parameterization options simulate rather scarce precipitation throughout the entire year (see Sect. 3.3 for more details on precipitation amounts). The maps of temporal correlation between PISCO and all other gridded data show negative correlations at the Pacific coast of Peru for all WRF setups, while IMERG and CHIRPS compare well to PISCO (Fig. 3f–l). This is expected

as the observation based data sets are not independent from each other.

The temporal correlation coefficients in the NE flatlands plotted as boxes and whiskers (Fig. 3e) show a rather homogeneous performance of the gridded observational data sets, where PISCO obtains the lowest median and the largest spread. This poor performance for PISCO is already pointed out by Aybar et al. (2020). Among the parameterization options, Kenya and Micro13 show the lowest medians of the correlation coefficients. However, only 15 stations were available in this large area,

which means that the evaluation of the parameterization options against the weather stations or the gridded observational data sets over the NE flatlands must be carried out with caution. In the NE flatlands, all parameterization options show a rather well correlated area in the southeastern part of the region, but this temporal correlation is lost towards the northwest (Fig. 3f, h, i, k and l). Kenya and Micro13 even show negative correlations in the northern part of the domain. However, Micro13 and South America are the only parameterization options that show good temporal correlations over Madre de Dios. A good temporal

correlation between the gridded observational data sets and PISCO is also highlighted in the flatlands (Fig. 3g and j). This good agreement between observational data sets is expected as they are not fully independent from each other, particularly in the regions where the number of available weather stations is small. This is particularly true for the Brazilian part of PISCO, as only stations from Peru are considered for the creation of PISCO. Consequently, this area fully depends on the information provided by the gridded observational data sets.

The plateau (Fig. 4a), the NE slopes (Fig. 4c) and the NE flatlands (Fig. 4e) are the three regions where the parameterization options show the largest RMSE against weather station data, particularly the Micro13 and South America simulations. It is noteworthy that PISCO shows a larger RMSE than IMERG and CHIRPS in the NE flatlands and particularly in the NE slopes. However, these are the regions where CHIRPS and IMERG are more precise than PISCO, as the median RMSE of PISCO against weather station data is higher and the spread is largest. This misrepresentation of PISCO might be related to the scarce

weather station availability and the respective miscorrection of the satellite data in that area. This might lead to a loss in the precipitation related to the complex atmospheric dynamics along the NE slopes and flatlands. The pattern of the RMSE against





PISCO is rather similar in all the parameterization options and gridded observational data sets, highlighting the largest values in the NE flatlands and slopes of the Andes (Fig. 4f–l). The lowest RMSE for the parameterization options are shown at the Pacific coast of Peru (Fig. 4d), which are similar to those obtained by PISCO. The RMSEs shown by ERA5 and the other
observational data sets are larger than the ones for the parameterization options. This agrees with the RMSE maps against PISCO, because precipitation amounts are very small during the entire year.

WRF is able to maintain its skill in terms of temporal correlations and RMSE also at finer temporal resolutions (not shown). The correlations and RMSEs of all parameterization options and gridded observational data sets show rather similar results for intervals of a months, 15-days and 10-days, independently of the region. For 5-day and daily intervals the values drop for the
correlations and rise for RMSEs. The increase in the RMSEs and the reduction in the correlations are expected due to the fact that capturing the exact amounts of precipitation at the same time as the observations is rather challenging for the model. The differences between the parameterization options, ERA5 and the gridded observational data sets considered are reproduced at finer temporal resolutions as seen in Figs. 3a–e and 4a–e.

In summary, the differences in parameterization options with respect to correlation coefficients and RMSE are rather small.
Nevertheless, Micro13 clearly stands out in the NE slopes, while it fails to capture the proper correlations in the SW slopes. In the SW flatlands Europe and South America show the best results, as they generally produce little precipitation amounts, which matches for the driest region of the country. While in the plateau the performance is overall very good, related also to the high station density, the opposite is true for the NE flatlands. Over Madre de Dios, only Micro13 and South America are able to keep high correlation values.

## 3.2 Pattern correlation analysis of precipitation

Since the temporal correlation analysis does not clearly constrain an optimal parameterization option over the entire region, we use the an additional measure, i.e., pattern correlation. The pattern correlations are calculated between the different parameterization options and the weather stations and the gridded observational data sets, separately. Due to the large RMSE and poor correlation of PISCO in the NE parts of the domain, which is in agreement with Aybar et al. (2020), CHIRPS is selected as
reference in those areas, while PISCO is considered for the plateau and the SW slopes and flatlands.

The monthly pattern correlations between the parameterization options and the reference data sets, i.e., PISCO or CHIRPS depending on the region, are shown in Figure 5. The pattern correlation value of 0.5 against stations and 0.4 against the gridded observational data sets are considered as moderate correlation values for each case, as they explain approximately a quarter of the variance (25 %). For the calculation of the pattern correlations against the weather station data, all the gridded observational
data sets are bi-linearly interpolated to the grid of the second domain of the WRF simulations using CDO. Conversely, for the pattern correlations against PISCO, the remaining gridded observational data sets and all the parameterization options are bi-linearly interpolated to the grid of PISCO. For the correlation against CHIRPS the same procedure is applied, but taking its grid as a reference for the interpolation.

The gridded observational data sets do not always agree well with the weather station data in terms of the spatial pattern
of monthly precipitation sums, particularly in the NE slopes of the Andes, the SW flatlands and the dry months in the SW





slopes. In the SW flatlands, only PISCO is able to capture correctly the precipitation pattern exhibited by the weather station data, while the other observational data sets show a poor performance from May to December. However, most of the monthly pattern correlations obtained in those regions are not significant. In the case of the parameterization options, they also show a rather poor performance in the NE slopes, particularly Europe and South America, which only exceed the threshold value of 0.5

in two months. In contrast to the observational data sets in the SW flatlands, the parameterization options show a rather good performance from May to December, but a poor performance in the first months of the year. Again, these pattern correlations are not significant. In the SW slopes, neither the different parameterization options nor the gridded observational data sets agree with the pattern of precipitation depicted by the weather stations during the dry season. The best performance of the parameterization options against weather station data is shown in the NE flatlands and the plateau, except for a few months of

the second semester and the first few months of the year, respectively.

Besides the pattern correlation analysis against the weather station data, the pattern correlations against PISCO and CHIRPS are calculated (second column of Fig. 5). All correlations are statistically significant (no starred pixels), and all gridded observational data sets agree relatively well with CHIRPS in the NE area of the domain, and with PISCO in the plateau. However, this is not the case for the SW regions against PISCO. In this region TRMM, IMERG and also ERA5 show negative correlation

coefficients from May to November, i.e., during the dry season and in the first months of the rainy season. The best performance of all sensitivity simulations is shown in the plateau, where nine to ten months obtain high correlation coefficients, except for South America, where only eight months show high correlation coefficients. It is noteworthy that the parameterization options fail to capture the correct patterns in January (one of the rainiest months of the rainy season). To test the effect of a possibly unbalanced land surface and to give the model more time to come into quasi-equilibrium, an additional sensitivity simulation

is performed using four instead of two months of spin-up. Nevertheless, no systematic improvement is found (not shown). As in the Pacific region of the domain some amelioration is observed, a somewhat longer spin-up period should be considered in the future. The performance of the parameterization options in the SW slopes against PISCO is similar to that shown against station data, with the best performance during the rainy months and the worst during the driest period. The best performance of WRF over the NE slopes is obtained by Micro13 since the threshold of 0.4 is exceeded in six months. The poorest perfor-

mance of the model is observed during the dry season. The other parameterization options are only exceeding the threshold in three or four months. The performance of the parameterization options is poorer over the NE flatlands, and particularly in the transition months between both seasons. Overall, Micro13 is the parameterization option with the best performance in the wettest and driest months (December to February, and July respectively). The worst performance of all the parameterization options is obtained at the Pacific coast of Peru, where zero months exceed the threshold of 0.4. This shows that WRF is not

able to simulate correctly the extremely dry conditions of the coastal area.

In summary, the pattern correlations show again that the plateau is relatively well captured by the model simulations and the gridded observational data sets. In the NE flatlands, the pattern correlation is rather good compared to the temporal correlation. In particular the Micro13 parameterization option performs well here. Precipitation patterns in both slopes are difficult to capture for the model compared to the weather station data, especially the NE slopes shows a poor correlation even with





PISCO. Parameterization options that perform best in the NE part of the domain are Micro13, and to some extent, Kenya and Europe. In the SW part of the domain No Cumulus outperforms the other parameterization options.

### 3.3  Precipitation patterns

Monthly precipitation patterns are further studied to evaluate the performance of the different WRF parameterization options. The accumulated precipitation maps for February — one of the wettest months during the rainy season — and for July —
the driest month of the dry season — are shown in figures 6 and 7, respectively. These two months were chosen as they are representative of the patters observed during the rainy and dry seasons, and because these two months in 2008 lie inside the inter-quartile range of the climatology for Madre de Dios (see Fig. 2b). Together with the maps, precipitation sums along a transect following ten weather stations through the Andes into Madre de Dios for both February and July are shown.

According to PISCO and CHIRPS, the largest precipitation amounts during February are observed in the NE slopes of the
Andes and towards the rainforest (Fig. 6). These maximum values in the slopes at the southern and eastern boarders of Madre de Dios are also observed in all the parameterization options. For the Europe parameterization option these peaks in precipitation are clearly underestimated compared to the other simulations and the gridded observational data sets. The largest differences between the parameterization options and both gridded observational data sets are found over the rainforest, particularly in the northern part of the domain. Most of the sensitivity simulations show rather dry conditions in that area, in contrast to
the comparably wet conditions depicted by PISCO and CHIRPS. Only the No Cumulus parameterization option does not show this feature in the northern part of the domain. However, No Cumulus shows a general excess of precipitation in the whole domain. Furthermore, the No Cumulus parameterization option invokes precipitation always at similar spots throughout domain 1 (25 km resolution), resulting in a rather patchy precipitation pattern in the outermost domain (not shown). Even though precipitation in domain 1 is generally underestimated, the No Cumulus parameterization option is able to (over)correct
this in domain 2. The transect figure shows that all the sensitivity experiments are able to follow the topography and produce precipitation according to it reasonably well. For PISCO and CHIRPS it becomes clear that their spatial resolution is not enough to follow correctly the topography of the mountains and to capture precipitation accordingly. This is highlighted particularly well at the fast change between the plateau and the flatlands in the NE Andes, as they show the precipitation peak over the flatlands and not in the slopes as it should be expected. Overall, PISCO and CHIRPS are able to reproduce well the observed
precipitation amounts of the stations. This is expected as information from weather station data is used to generate these data sets. Except for the Micro13 parameterization option, most of the simulations exceed the measured values of precipitation in the weather stations of the SW flank of the Andes and the plateau. As discussed above, Micro13 is the only parameterization option that captures high enough precipitation amounts with respect to the weather station data in the NE flatlands compared to the other parameterization options.

During July, the precipitation over the entire domain is rather scarce according to PISCO, and only in some local parts of the domain the value of 25 mm per month is exceeded (Fig. 7). In the NE part of the domain, CHIRPS — the preferred data set over that region — shows wetter conditions. Most of the parameterization options agree with PISCO and CHIRPS in the absence of precipitation in the SW slopes of the Andes and the SW flatlands. For the plateau, the simulations agree with PISCO





on the rather dry conditions, except for Micro13 and CHIRPS that show wetter patterns. The main differences in precipitation

amounts between both observational data sets and the different parameterization options are identified in the NE slopes of the Andes. However, the maxima are located in the same areas as in CHIRPS. Micro13 is the wettest simulation in that area, followed by Kenya, South America and No Cumulus simulations. Note that only Micro13 is able to produce similar patterns and amounts of precipitation in the flatlands compared to PISCO, which in this month, shows the highest pattern correlations against weather station data (Fig. 5). The wide agreement of the Micro13 simulation with the weather station data in the NE

flatlands is also highlighted by the high pattern correlation in Fig. 5. As in February, Europe is again too dry over the entire domain compared to PISCO and especially CHIRPS. Similar results are shown in the transect through the Andes towards Madre de Dios. There, the excess of precipitation of Micro13 is highlighted, and also the effect of a finer representation of the topography in precipitation for the parameterization options compared to the observational data sets. Again, both CHIRPS and PISCO show the precipitation peak over the lower part of the slopes or the flatlands.

Altogether, the parameterization options show different behaviours over the entire domain and seasons. In the wet season the Europe parameterization option is too dry, which applies to some extent to Kenya as well, particularly in the rainforest. Micro13, South America and No Cumulus provide reasonable patterns and amounts for the rainy season, except for the last which overestimates the amounts. As expected, the South America parameterization option simulates precipitation reasonably in the NE slopes of the Andes, since this configuration is designed to capture storms in this area. In the dry season, all the

parameterization options simulate too dry conditions. Micro13 must be excluded from this, as its precipitation patterns and amounts are similar to the ones shown by PISCO and CHIRPS, rendering Micro13 as the optimal setting for Madre de Dios.

### 3.4    Seasonal cycle over the northeastern flatlands

To investigate the reason for the different performances of the parameterization options over the northeastern flatlands, which covers the region of interest, field means of different variables are analyzed by means of a seasonal cycle and daily cycles.

The Europe parameterization option simulates especially low monthly precipitation sums in the wet seasons, while the other parameterization options show almost no precipitation during the dry season (Fig. 8a). The Micro13 parameterization option simulates more precipitation than the others in all months as already shown in Fig. 6 and 7. The same is also true for the precipitable water, i.e., the vertically integrated water vapour content of the whole atmospheric column. Here, the difference between the Micro13 parameterization option and the other options is evident in the first half of the year (Fig. 8c). The

combination of cooler 2-meter temperatures (Fig. 8b) and a higher water vapour content results in a higher relative humidity at 2-meters, especially in July, but also during the first half of the year. Even though the average 2-meter temperature of the Europe option is similarly low as in Micro13, the relative humidity is not higher compared to the others, even if the water vapour content is similar. The reason for this might lay in the daily cycle of 2-meter temperature (Fig. 9). While the monthly average is similar to the Micro13 option, the daily cycle looks completely different. The temperature is lower than for the Micro13 option for

Europe, while during the day the temperature is much higher and hence, the relative humidity decreases on average compared to the Micro13 option. The relative humidity and the precipitable water of the No Cumulus parameterization option is especially low, even though the precipitation is comparable to the other options, which might indicate that this parameterization option has





an efficient process to remove moisture from the atmosphere, i.e., convective processes. This is also supported by the fact that precipitation occurs mainly in the afternoon, while the other options have precipitation distributed over the whole day (Fig. 9).

A striking difference between the Micro13 and the other parameterization options is the fraction of mid-level clouds and the soil moisture content. A high cloud cover allows for more precipitation, but also to reduce atmospheric temperature, which is especially true for a high fraction of high-level clouds (not shown). The difference in soil moisture is induced by the higher precipitation amounts, which then allows to fuel the water vapour content in the atmosphere, leading to more precipitation and supporting a positive feedback. Hence, for the simulation of the convection permitting domain, the change in the microphysics

parameterization results in the largest changes of precipitation and related processes.

### 3.5 Performance under wetter climate conditions: year 2012

To evaluate the applicability of the best parameterization options to other years showing different climate conditions, i.e., during the rather wet year 2012, further experiments were run. The Europe experiment was excluded as the analysis of the year 2008 already shows that this option is too dry during the entire year. The temporal correlation of monthly precipitation

sums against weather station data shows that, in general terms, the performance of the parameterization options under wetter conditions is rather similar to that of year 2008 (Fig. S1a). The overall performance of the parameterization options in terms of temporal correlations is improved in the SW slopes, the plateau and the NE flatlands, while the performance is slightly worse in the SW flatlands and the NE slopes. The RMSE of all the different parameterization options and observational gridded data sets is slightly larger than in 2008 in the NE parts of the domain (Fig. S1b). CHIRPS should be still preferred as a reference in the

NE slopes of the Andes in 2012, but not anymore in the NE flatlands since PISCO shows higher correlations and lower RMSEs. The poor agreement between the gridded observational data sets and weather station data in terms of the spatial patterns of monthly precipitation sums in the NE slopes and the SW flatlands is also observable in 2012 (Fig. S2). Micro13 and Kenya still remain the best parameterization options for the NE part of the domain in 2012. The same is true for the No Cumulus option in the SW part of the Andes.

Micro13 captures the precipitation patterns relatively well in both rainy and dry seasons of 2012. In the rainy season, i.e., in February (Fig. S3a), Micro13 is a bit too dry in the northern part of the domain. Kenya shows a larger deficit of precipitation over the same area. Note that February 2012 is an anomalously wet month compared to the climatology of 1981–2010 (see Fig. 2), which highlights the ability of Micro13 to even simulate correctly anomalous monthly precipitation amounts. During the dry season, i.e., in July (Fig. S3b), Kenya and No Cumulus show rather dry conditions over the Amazon, and only Micro13

shows comparable amounts to those in PISCO (and CHIRPS – not shown).

### 4 Summary and Conclusions

This study aims at determining the optimum setup for WRF to accurately simulate the observed precipitation patterns and amounts over the Amazon basin in southeastern Peru. The region of interest is the entire department of Madre de Dios, but because of the lack of a dense network of weather stations in that area we evaluate the performance of the model over a broader





area including the tri-national border of Peru, Bolivia and Brazil. Besides the scarce availability of weather station data also the large-scale atmospheric circulation controls and the complex topography of the region renders this part of the world a challenge for regional climate models. The novelty of this study is that this is one of the first times that such a complex region as southern Peru is resolved at a convection permitting spatial resolution down to 1 km, and consequently, several physics parameterizations must be tested in the regional climate model, which is WRF for this study. We tested different combinations of microphysics (WSM6 and SBU), cumulus (Grell-Freitas, Kain-Fritsch or no cumulus), LW (CAM and RRTM) and PBL (ACM2 and YSU) parameterizations in a total of five different simulations. For the analysis, the year 2008 is selected since the climatology of precipitation over Madre de Dios suggests that it is more or less a standard year. Even though this year is starting during a strong La Niña event, which weakens in the middle of the year and starts to emerge again at the end of the year, the impact of ENSO on Madre de Dios in terms of precipitation is considered to be limited. This is particularly true for the rainy season at the beginning and end of the year, where La Niña phases are observed. Apart from weather station data obtained from WMO, data from Peru, Bolivia and Brazil were also provided by SENAMHI, SENAMHI Bolivia and INMET, respectively. Additional gridded observational data sets are employed in our study to validate the sensitivity simulations. These include: PISCO, CHIRPS, IMERG and TRMM.

First, the gridded observational data sets are compared to the weather station data. The following characteristics for the gridded observational data sets over southern Peru are identified:

- The temporal analysis over the monthly data shows that the agreement between the gridded observational data sets and the weather station data is high. This is true for both evaluated indices, Spearman correlation and RMSE. In the case of the ERA5 reanalysis, this is only true for the Spearman correlations, as its RMSEs are larger than the ones from the observational gridded data sets.

- The monthly pattern correlation analysis shows that all the observational data sets perform poorly over the NE slopes of the Andes, and also in the SW slopes during the dry months. Additionally, the pattern correlations of TRMM, IMERG and CHIRPS are deficient at the Pacific coast from May onward.

- The quality and the station density of the rain gauge measurements over the plateau is high enough to validate the sensitivity simulations, but not necessarily in the remaining regions.

- The lack of a dense weather station network over the Amazon reduces the quality of PISCO in the NE slopes and flatlands. Instead, CHIRPS should be preferred in these regions, which is in line with Aybar et al. (2020).

After evaluating the weather station and gridded observational data and identifying which data set must be used as reference in which region, the WRF simulations are then tested. By comparing the WRF simulations to the chosen data sets the following conclusions are drawn:

- The temporal correlations and RMSEs of the sensitivity simulations show that the best correlations and lowest RMSEs are obtained over the plateau for all the WRF simulations. The poorest correlations are shown at the Pacific coast of Peru, while the largest RMSEs are found in the NE slopes.





- The largest differences among the sensitivity simulations in the temporal correlations are seen in the NE flatlands. Only Micro13 and South America are able to keep high temporal correlation values over Madre de Dios, while Micro13 and Kenya perform best spatially.

- The monthly pattern correlation exhibits the poorest performance in the coastal region of Peru. Even though the general pattern is not well captured by the simulations, the scarce precipitation amounts are correctly simulated locally. The first months stand out as particularly poor which also applies to the Amazonian flatlands and the plateau, which supports the idea that a longer spin-up is needed. Nevertheless, no systematic improvements were observed when four months instead of two where employed as spin-up (not shown).

- The precipitation patterns in the wet season are captured reasonably well by all parameterization options, but the amounts are clearly underestimated by the Europe option and overestimated by No Cumulus. Kenya shows an underestimation in the NE flatlands, while Micro13 simulates too much precipitation in the NE slopes. South America performs well, as this option is designed to capture storms in the NE part of the domain.

- For the dry season, most of the parameterization options simulate too dry conditions. This includes the Europe, South America, Kenya and No Cumulus parameterization options. The last two are able to correct the dryness in the NE flatland by some extent. Micro13 provides good precipitation results on the whole domain, except for an overestimation in the NE slopes.

- The change in the microphysics parameterization from the WRF single-moment 6-class scheme to the Stony-Brook University scheme causes a great improvement in the representation of precipitation over the Amazon, and consequently, reinforces a positive feedback between the soil moisture, temperature, relative humidity, cloud cover and finally, precipitation.

- These results indicate that the configuration over Kenya is not optimal for another equatorial region as Peru, and supports the fact that parameterization options are not fully interchangeable, as also found for example by Takle et al. (2007), Jacob et al. (2012) and Russo et al. (2020).

- The transects through the Andes into the rainforest of Madre de Dios suggest that the spatial resolution of WRF is necessary to capture correctly the precipitation associated to the terrain, while this is not the case for PISCO or CHIRPS. Thus, the biases in the slopes of Micro13, particularly against PISCO, may be related to the lack of high spatial resolution in this complex region.

Hence, we conclude that Micro13 should be the preferred setting for southeastern Peru in WRFV3.8.1, and particularly for the area of Madre de Dios. Similar conclusions are obtained from the analysis performed for the wet year 2012. Thus, the robustness of the results presented here are highlighted, since this WRF model setup is not only applicable to standard years in terms of precipitation, but also to other climate conditions.

This methodological study is a starting point to gain a deeper understanding of the individual drivers over this complex
region. It further reaffirms the need for the selection of parameterization options for different regions in southern Peru, as
it seems rather difficult for the model to perform well in all of the five regions analyzed here. This study can foster new
developments of microphysics parameterization schemes in WRF. Furthermore, different climate simulations can be carried
out to evaluate the effect of global warming, but also to investigate land-atmosphere interactions. This is particularly important,
as this region strongly depends on natural resources, yet it suffers from various anthropogenic influences, such as deforestation,
wild fires and mining. To conserve the biodiversity capital of Peru it is important to understand the impact of different drivers
of change including the regional impact of anthropogenic climate change in particular.

*Code and data availability.*    The Weather Research and Forecasting (WRF) model V3.8.1 can be downloaded from the user's page online
(https://www2.mmm.ucar.edu/wrf/users/download/get_sources.html, last access: 9 November 2021). All the postprocessed outputs from all
the sensitivity simulations performed with WRF and included in this manuscript are available in: https://doi.org/10.5281/zenodo.5378127.
The namelist files used to run each experiment, and the programming codes employed in the analysis are also included in the previous link to
Zenodo. The gridded precipitation data sets included in our research can be found online. The Earth Observing System Data and Information
System (EOSDIS) provides access to TRMM and IMERG data sets (https://doi.org/10.5067/TRMM/TMPA/3H-E/7 and https://doi.org/10.
5067/GPM/IMERG/3B-HH/06, respectively), while the Climate Hazards Center of the University of California Santa Barbara holds the data
for CHIRPS (https://www.chc.ucsb.edu/data/chirps, last access: 11 November 2021). PISCO can be downloaded from the IRI Data Library
(https://iridl.ldeo.columbia.edu/SOURCES/.SENAMHI/.HSR/.PISCO/.Prec/.v2p1/?Set-Language=en, last access: 11 November 2021). The
weather station data from Peru was provided by SENAMHI, while the data from Bolivia and Brazil was downloaded from SENAMHI
Bolivia (http://senamhi.gob.bo/index.php/sismet, last access: 10 October 2021) and INMET (https://bdmep.inmet.gov.br/, last access: 10
October 2021). The additional stations obtained from WMO were obtained from the World Weather Records website (https://www.ncei.
noaa.gov/access/search/data-search/global-summary-of-the-day, last access: 13 October 2021).

*Author contributions.*    The idea behind this project was developed by all the authors. The preparation of the data sets and the design of the
methodology was carried by SJGR and MM. All the authors took part in the analysis of the simulations. The original draft was written by
SJGR and MM, but all authors contributed with their knowledge and ideas to the manuscript.

*Competing interests.*    The authors declare no competing interests.

*Acknowledgements.*    This research was financially supported by the the Oeschger Centre for Climate Change Research and the Wyss Founda-
tion through the pilot project of the Wyss Academy for Nature at the University of Bern. Thomas F. Stocker and Christoph C. Raible received
financial support by the Schweizerischer Nationalfonds zur Förderung der Wissenschaftlichen Forschung (grants no. 200020_172745 and
200020_200492). This work was supported by a grant from the Swiss National Supercomputing Centre (CSCS) under project ID s905. The



authors acknowledge SENAMHI for sharing part of the weather station data of Peru for year 2008 with us. The TMPA data were provided by the NASA/Goddard Space Flight Center's Mesoscale Atmospheric Processes Laboratory and PPS, which develop and compute the TMPA as

a contribution to TRMM. Finally, part of the analysis was performed using R (R Core Team, 2021), and the authors acknowledge the creators of the packages used for it.





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



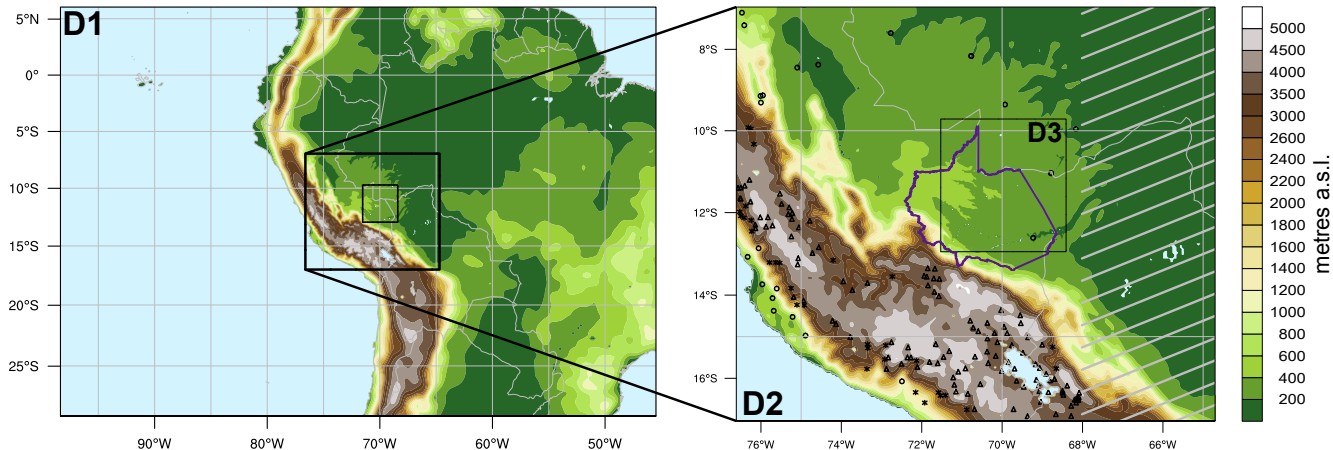

**Figure 1.** The three nested domain setup employed in all the WRF experiments is depicted on the left. A nesting ratio of 1:5 is used, which means that the spatial resolution of the largest domain is 25 km, followed by another domain with 5 km spatial resolution, and the innermost domain focusing over the region of Madre de Dios with 1 km horizontal resolution. The colour shading indicates elevation in metres above sea level (a.s.l.) using the WRF topography Global Multi-resolution Terrain Elevation Data (GMTED2010) provided by USGS. Domain 2 is presented in detail on the right, where the weather stations for the year 2008 are presented. The stations in the flatlands (< 1000 metres a.s.l.) are depicted with a circle, the ones in the slopes of the Andes (1000–3000 metres a.s.l.) with an asterisks, and finally, stations in the plateau (> 3000 metres a.s.l.) are marked as triangles. The grey striped area highlights the region that is not included in the analysis against PISCO, since this data set is only available over a rectangle around Peru. The purple line boarders the department of Madre de Dios.



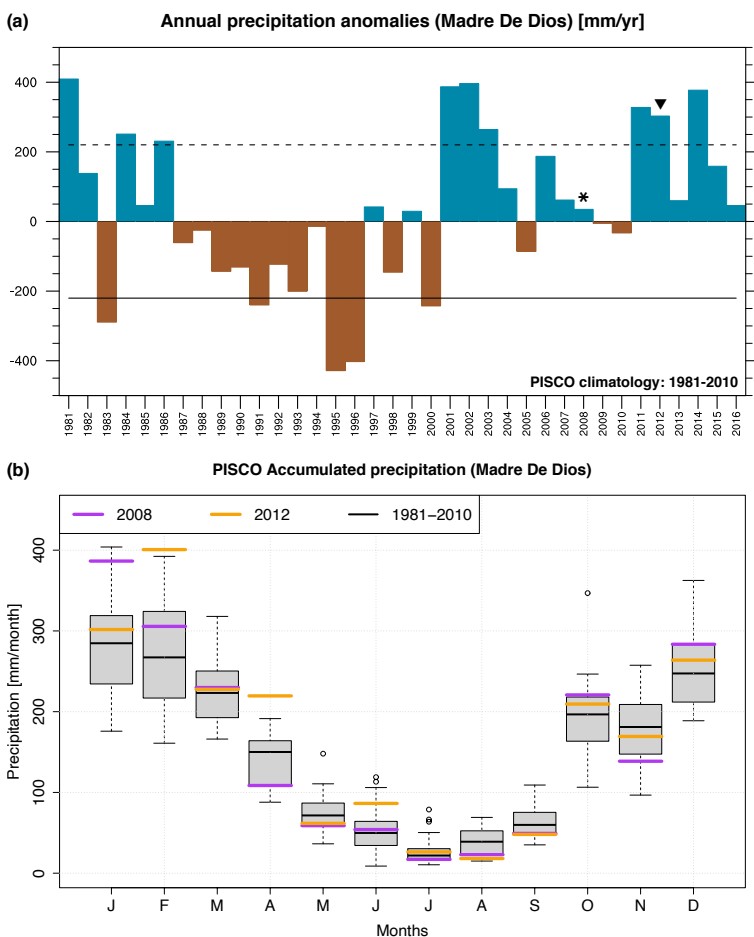

**Figure 2.** (a) Annual precipitation anomalies (in millimetres per year), where the asterisks denotes our main study year 2008 and the triangle the wet year 2012, and (b) monthly accumulated values of precipitation (in millimetres per month) for years 2008 and 2012 (purple and orange horizontal lines, respectively) compared to the climatology (1981–2010, in grey, using a box and whisker plot). All values are means of the PISCO gridded data set over the department of Madre de Dios (check Fig. 1 for exact location of Madre de Dios). The stippled and straight lines in (a) illustrate plus and minus one standard deviation, respectively. The whiskers in (b) extend to the value that is no more than 1.5 times the inter-quartile range away from the box. The values outside this range are defined as outliers and are plotted with dots.



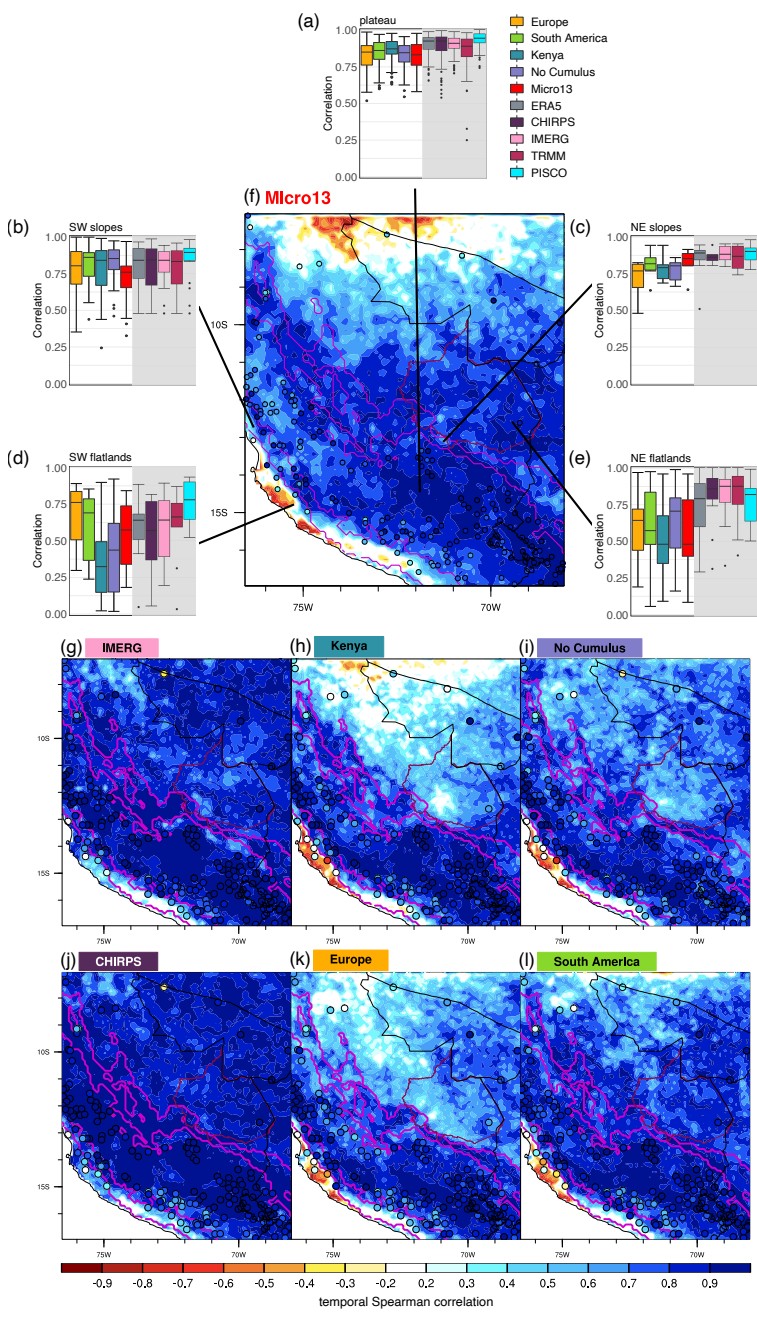

**Figure 3.** Temporal Spearman correlation against weather station data in the box and whisker plots (a–e) and against PISCO data set (f–l). The box and whiskers are divided into five groups according to the elevation of each station and location related to the Andean mountain range: (a) the plateau, (b) the SW slopes, (c) the NE slopes, (d) the SW flatlands and (e) the NE flatlands. The circles on the map indicate the Spearman correlation coefficient with respect to the corresponding weather station. Note that the WRF domain is reduced at the eastern side to match the PISCO data set (refer to Fig. 1 for more details). The grey shading in panels (a) to (e) denotes the satellite based or reanalysis data, and facilitates the separation from the WRF simulations.

10.5194/gmd-2021-307
Geoscientific Model Development
2021-11-23




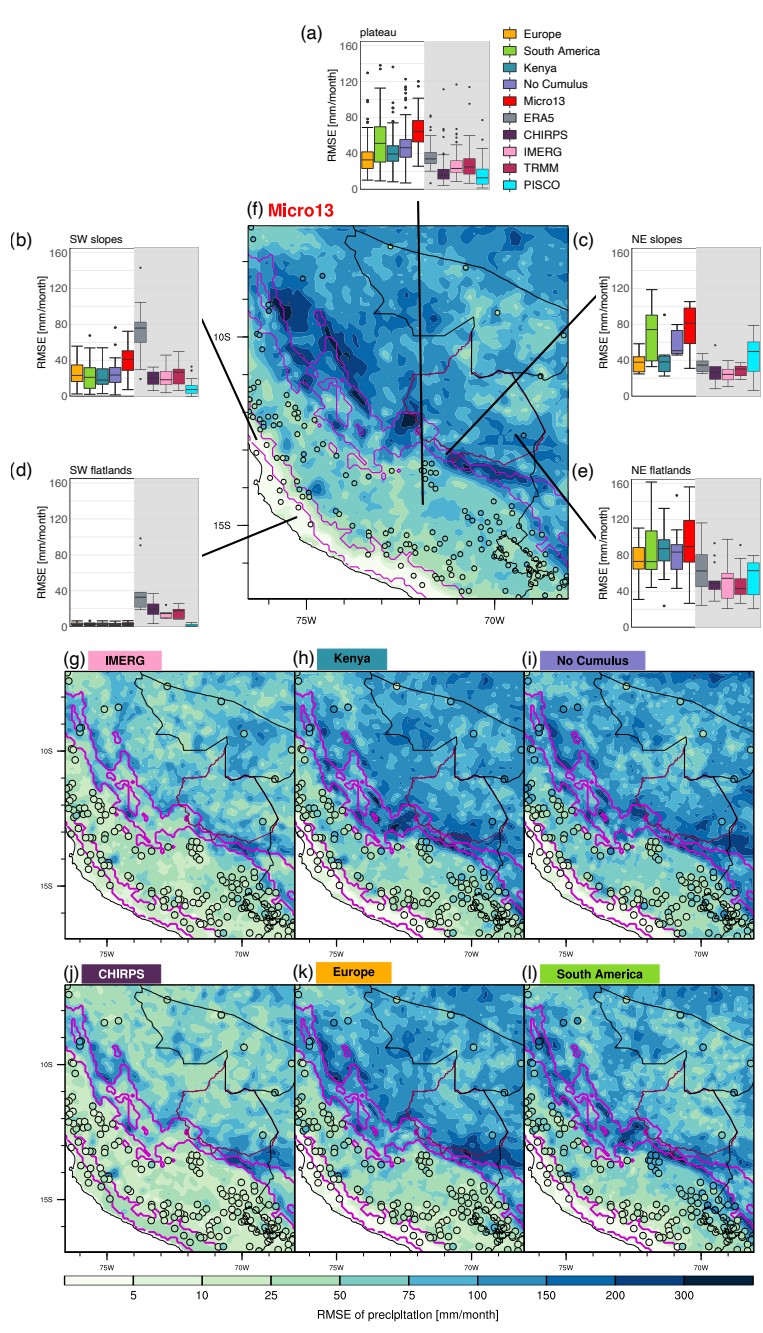

**Figure 4.** Same as Figure 3 but for RMSE (in millimetres per month).



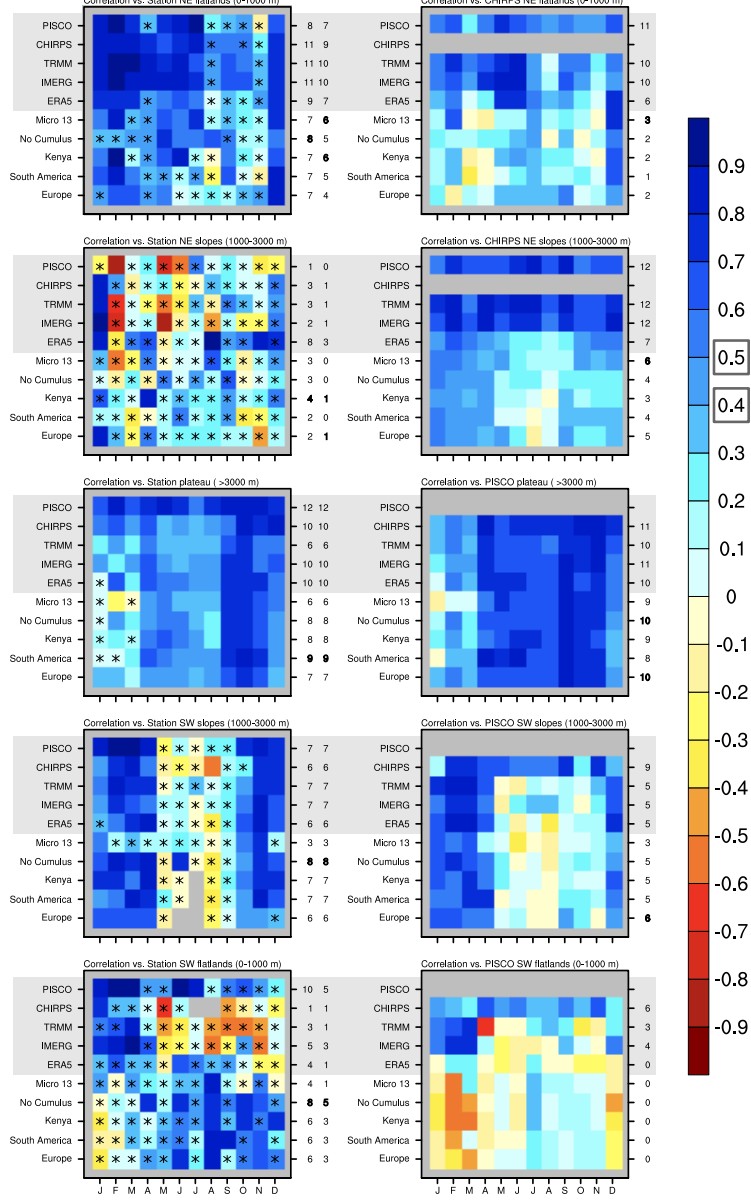

**Figure 5.** Spearman pattern correlation compared to weather stations data (left column) and compared to CHIRPS (first two rows, right column) and PISCO (last three rows, right column). The five rows represent stations and data points in the northeastern flatlands (< 1000 metres a.s.l.), along the northeastern slopes of the Andes (1000–3000 metres a.s.l.), on the plateau (> 3000 metres a.s.l.), along the southwestern slopes of the Andes (1000–3000 metres a.s.l.), and in the southwestern flatlands (< 1000 metres a.s.l.), respectively. The light grey shading denotes satellite based or reanalysis data, and facilitates the separation from the WRF simulations. Asterisks inside the pixels indicate non significance at $\alpha = 5\,\%$. The numbers in the first column of the right y-axis indicates the number of months that result in a correlation larger than 0.5 compared to weather station data (left panels) and 0.4 in comparison to PISCO data (right panels). The second column of the right y-axis considers only months that are statistically significant (left panels only). The bold numbers indicate the best option for each region.





**Figure 6.** Monthly precipitation sums in millimetres per month for February 2008, a month in the rainy season, are shown for the parameterization options Micro13, Kenya, No Cumulus, Europe and South America and the gridded observational data sets PISCO and CHIRPS. The circles on the map indicate the monthly precipitation sums recorded at the respective weather station. Grey dots indicate missing values at the respective station. Monthly precipitation sums along a transect through the Andes into Madre de Dios are also shown. The transect follows the purple line on the inset in the upper right corner of the plot. The dots on the inset and the symbols below the $x$-axis mark the location of the weather stations. Precipitation measures from all the sensitivity simulations (full lines) and the reference data sets PISCO and CHIRPS (stippled lines) are depicted, while the observed precipitation amounts of each station along the transect are included as bars. The grey shading indicates the topography along this transect given in kilometres a.s.l. (right $y$-axis). The separation into flatlands, slopes and plateau is also given at the right side of the transect panel.





**Figure 7.** Same as Fig. 6, but for July, which represents the dry season.



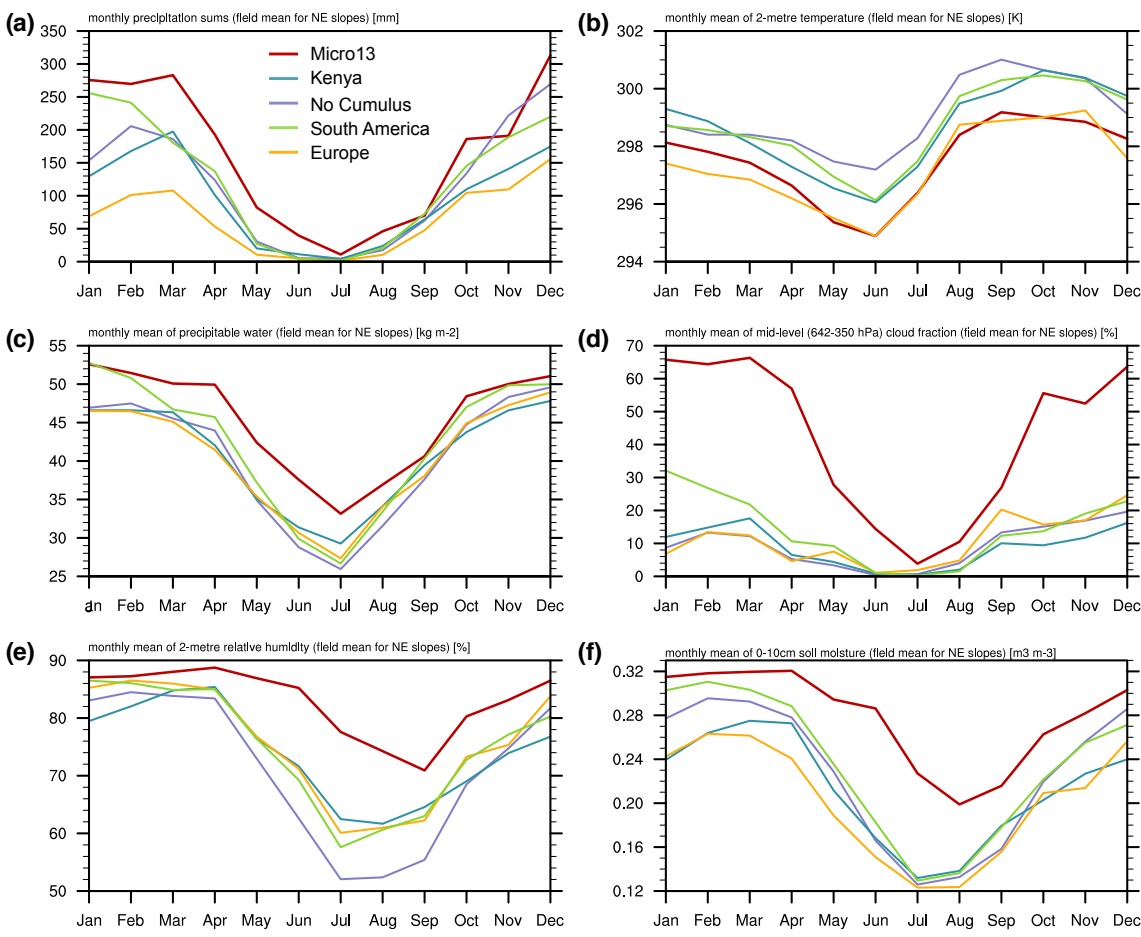

**Figure 8.** Seasonal cycle of a field mean over the northeastern flatlands for (a) monthly precipitation sums (mm), (b) monthly mean of 2-metre temperature (K), (c) monthly mean of precipitable water (kg m$^{-2}$), (d) monthly mean fraction of mid-level (642–350 hPa) clouds (%), (e) monthly mean of 2-meter relative humidity (%), and (f) monthly mean soil moisture in the first soil layer (0-10 cm, m$^3$ m$^{-3}$). The $x$-axis shows the months of the year.



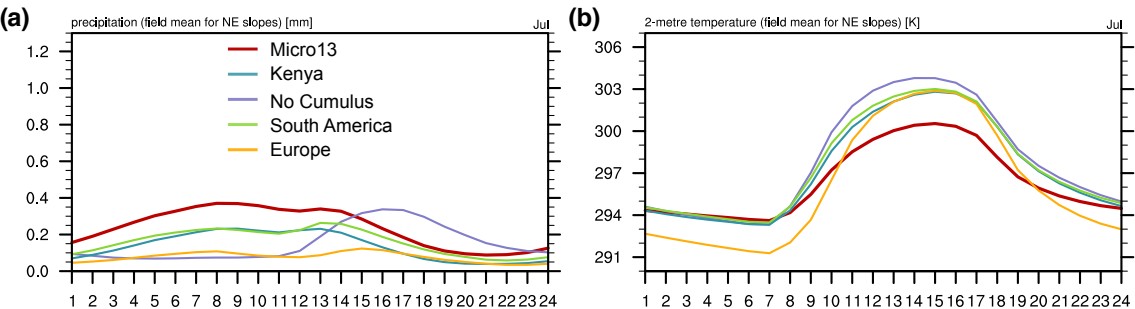

**Figure 9.** Daily cycle for July of a field mean over the northeastern flatlands for (a) precipitation (mm), and (b) 2-metre temperature (K). The $x$-axis shows the hours of the day, shifted by 5 hours from UTC to show local Peru time.





**Table 1.** The setups of each simulation are described with: name of the experiment, parameterizations used and other parameters, such as nesting option and the number of domains. Bold domains indicate that the cumulus parameterization is turned on.

| Name | Parameterizations | | | | Other parameters | |
| --- | --- | --- | --- | --- | --- | --- |
| | Micro | Cumulus | LW-Rad. | PBL | Nesting | Doms in Fig. 1 |
| Europe | WSM6[1] | Grell-Freitas | CAM[2] | ACM2[3] | two-way | **D1**, D2, D3 |
| South America | WSM6 | Kain-Fritsch | RRTM[4] | YSU[5] | two-way | **D1**, D2, D3 |
| Kenya | WSM6 | Grell-Freitas | RRTM | YSU | one-way | **D1**, D2, D3 |
| Micro13 | SBU[6] | Grell-Freitas | RRTM | YSU | one-way | **D1**, D2, D3 |
| No Cumulus | WSM6 | - | RRTM | YSU | one-way | D1, D2, D3 |

[1] WRF single-moment 6-class scheme. [2] Community Atmosphere Model. [3] Asymmetric Convection Model Version 2. [4] Rapid Radiative Transfer Model. [5] Yonsei University. [6] Stony–Brook University.