# Peer review of "Sensitivity of precipitation in the highlands and lowlands of Peru to physics parameterization options in WRFV3.8.1"

_Geoscientific Model Development, 2021_

## Referee Comment (RC2)

**Title:** Sensitivity of precipitation in the highlinds and lowlands of Peru to physics parameterizations options in WRF3.8.1

**Authors:** Santos J. González-Rojí, Martina Messmer, Cristoph C. Raible, and Thomas F. Stocker

**Reference:** gmd-2021-307

**Journal:** Geoscientific Model Development

This work evaluates the performance of the Weather Research & Forecasting (WRF) model version 3.8.1 at convection-permitting scale over southern Peru using different configurations combining microphysics, cumulus, longwave radiations, and planetary boundary layer physic schemes. For this purpose, different comparisons for two years, 2008 and 2012, were performed between the WRF outputs and the observations of both weather stations and precipitation gridded products.

The topic addressed in this study is relevant, being the results here found of high value for the climate scientific community. The manuscript is well written and structured, with an appropriate discussion of the results, showing clear and concise conclusions. Therefore, I recommend this study for publication in the Geoscientific Model Development (GMD) journal after minor revisions. My comments are as follows:

**Major comments:**

1. The main analysis was based on 2008 as the precipitation over Madre the Dios was more or less standard (L437-438) in that year. In addition, the year 2012 was selected to test the model performance for wetter conditions. In this regard, as analysis focused on domain 2, how were 2008 and 2012 for the whole d02 domain? Were they also "standard" and "wet"? To clarify this point, it could be good to represent the climatology of domain 2 in Figure 2.

2. For me, one of the most interesting parts of this study is the one related to section 3.4 (seasonal cycle over the northeastern flatland). For these analyses, a comparison could be made with observational values (for example, ERA5) to see which combination of parameterizations are closer to the " reality"? On the other hand, and just as a curiosity, why did the authors select IMERG and CHIRPS (and not the other two, i.e., ERA5 and TRMM) to compare with PISCO?

3. Although the analyses proposed here at the monthly scale provide very valuable information on the characterization of the model and the different parameterization schemes, the use of the daily scale could provide additional information on how the model represents extreme values at a high spatial resolution, and which scheme is better in this regard for the different regions.

4. If I understood correctly, the results of domain 3 were not used for the evaluation of the model performance. I agree with the authors to focus the evaluation on domain d02, since, as mentioned by the authors in L428-430, there are not enough observations for a comparison at the finest spatial resolution (i.e., 1 km). However, due to the high computational cost required to run the model for an additional domain at 1-km, it might be good to better specify what was the final purpose of using a third domain, covering the department of Madre de Dios, in such a high spatial resolution.

**Minor comments:**

- L121 "*...so a spin-up period of two months is enough to balance the fluxes between the atmosphere and soil in WRF*": Here, maybe, I would not affirm that a 2-month spin-up period is enough since, as the authors concluded, a longer spin-up period is probably needed for the simulations (L315-318 and L467-469).
- In section 2.1., please specify the number of vertical levels used in WRF, the top of the model, and the time-steps applied in simulation.
- Please provide information about the time resolution of the weather station data in the main text.
- L211: Did the authors check other interpolation methods? Please justify why bilinear interpolation was used instead of others (e.g., nearest neighbor).
- L327 "*In the NE flatlands, the pattern correlation is rather good compared to the temporal correlation*": Here, it could be good to remember that the comparison is between the results from Figure 5 and Figure 3.
- L346-347 "*However, No Cumulus shows a general excess of precipitation in the whole domain*" and L378 "*... except for the last which overestimates the amount*": It is hard for me to see that the No Cumulus combination generates more precipitation, in general, for the entire domain than others parameterization schemes (e.g., Micro13). In this regard, it could be good to indicate the mean value of the accumulated monthly precipitation for the entire domain, or even the mean for the different five regions.
- L356-359 "*Except for the Micro13 parameterization option, most of the simulations ...in the NE flatlands compared to the other parameterization options*"*:* Were the authors referring to the results obtained from the transect? Please, clarify this point.
- L363-364 "*For the plateau, the simulations agree with PISCO on the rather dry conditions, except for Micro13 and CHIRPS that show wetter patterns*": Here, I would suggest removing the information for CHIRPS as the comparison seems to be between simulations and PISCO. Otherwise, I would change the sentence to express it in another way.

**Figures**

- Figure 3a-e: I would suggest changing the color of the box for CHIRPS. Here, the median is sometimes hard to see.
- Figure 3f-l: I would suggest changing the colors of the lines bordering the different regions. It is sometimes difficult to differentiate between the borders of the regions (i.e., plateau, SW slopes, SW flatlands, NE slopes, and NE flatlands) and the borders of Madre de Dios. Also, if not necessary, it could be good to remove the black lines in all the maps. Do these lines represent the borders between countries?
- Figure 4b: I would suggest changing the range on the y-axis for this case to better show the box and whisker plot.
- Figures 6 and 7: I would suggest adding the borders between the five regions (as in Figures 3 and 4) in order to better follow the discussion of the results.

---

## Author Comment (AC1)

**Point-by-point Reply to Referee #1**

**Comment published online on the 20th of December, 2021**

*In their work, González-Rojí et al. present an assessment of the WRF model skill at simulating precipitation over parts of South America, with a focus on parts of Peru, Brazil and Bolivia. The authors explore different configurations of the WRF model, evaluating the performance of the model by comparing precipitation outputs to observations from weather stations and estimates from observation-based products and ERA5. From their analysis, the authors are able to identify which of the studied model configurations work best for their region of interest. In addition, the authors identify some strengths and weaknesses from observation-based products, like PISCO and CHIRPS.*

*The manuscript is well written and the topic is very relevant both for the climate science community of South America and the convection-permitting modeling community. I recommend this study for publication after minor revisions. Please see the details below.*

Thank you for reading our manuscript so carefully and for your positive and constructive comments.

*Major comments:*

*1. Taking further advantage of the high resolution simulations:*

*Simulations at such high resolution are very valuable for the region of interest. The analysis of the monthly accumulated precipitation is very interesting. A further analysis of 5-day or daily accumulated precipitation would help to strengthen the paper. For example, the analysis of the statistical distributions of the daily accumulated precipitation would help to identify the value of high resolution simulations at representing extreme precipitation.*

As stated in the manuscript, the temporal analysis at finer temporal resolutions such as 15-days, 10-days, pentads or daily was also carried out for our analysis, and the same results are observed for the different intervals. In general, the RMSE increases and the temporal correlations decrease as we increase the temporal resolution. The statement in the manuscript is based on figure R1.1, that is neither shown in the manuscript nor in the supplementary material. Hence, we will add this figure to the supplementary material.

[Figure]

*Figure R1.1:* (a) *The temporal correlation and* (b) *root-mean-square error (RMSE) between the annual cycle for the year 2008 of measured and simulated daily precipitation sums at the nearest grid point to the station's location shown for the different parameterization options and gridded observational datasets. The whiskers extend to the value that is no more than 1.5 times the inter-quartile range away from the box. The values outside this range are defined as outliers and are plotted with dots.*

*In addition, in the last section about the diurnal cycle, it would be interesting to add the observations, if hourly data is available.*

We agree that it would be interesting to include the observations to the plot, but the weather station data obtained for the validation of the sensitivity experiments is only available with daily temporal resolution. For precipitation we have added the data of IMERG (Fig. R1.2), as it is available with 30 min temporal resolution, but it must be noted that IMERG is not the best gridded data set available in any of the five regions studied. For all the other variables we are not aware of any data set with higher than daily temporal resolution. Figure R1.2 shows that Micro13, South America and Kenya are able to capture relatively well the precipitation of the first half of the day, but they all miss the peak in precipitation in the afternoon. Conversely, NoCumulus is able to simulate a peak of precipitation during the afternoon. We will replace this panel in the new version of the manuscript.

[Figure]

*Figure R1.2: Monthly mean daily cycle for July of a field mean over the northeastern flatlands for precipitation (mm) including also IMERG (pink line).*

*2. More details about the configuration and domains:*

*Include some other standard details about the simulations, like the number of vertical levels, model top, type of nudging (if used). In addition, please be more explicit about which of the domains is used in each part of the paper (for example state explicitly whether results from D02 or D03 are used in section 3.4, and so forth).*

All the sensitivity simulations include 49 vertical eta levels until the model top at 50 hPa, and the adaptive time step was employed while running the simulations. No nudging was applied to the input data. These details will be added to section 2.1 as suggested. Additionally, we will be more specific about the fact that we only show results from the second domain (D02).

*3. A small extension of the relationship between variables:*

*Both figures 8 and 9 are very interesting, as they allow to talk about possible relations between variables looking for an explanation about the behaviour of the simulated precipitation. It would be very interesting to see this analysis extended to at least one of the other regions studied in Figure 3. The authors could select for comparison, for example, the regions in flatlands vs. regions over the plateau, or over the slopes, where the simulated cloud field (both in terms of magnitude of cloud fraction and periodicities -e.g. annual and diurnal cycle-) might be qualitatively different.*

The main reason why we only evaluated the seasonal and daily cycles of different variables over the northeastern flatlands was because our region of interest (the department of Madre de Dios) is located there. It is true that the cloud cover is different depending on the region, and we can repeat the analysis for the plateau. Depending on the outcome of this analysis we will consider including or not an additional figure to the supplementary material, as the manuscript is already quite long.

*4. A comment about the order of the sections:*

*The manuscript is well written, and the sections are clear. However, it seems to me more clear to start with section 3.3, where the mean biases are presented, and then go to sections 3.1 (temporal correlations) and 3.2 (spatial correlations) where second-order metrics (correlations and RMSEs) are studied.*

We started analyzing the RMSEs and the temporal and spatial correlations as they provide a quantitative way to evaluate the performance of the simulations compared against weather stations and gridded observational data sets. However, it is true that we can also start section 3 by showing the accumulated precipitation maps first, and then continue with the more quantitative analysis of the results. We will consider adapting the structure of the paper if the consistency or the story line are not affected by this change.

*Minor comments:*

*L147-149. The authors write:*

*"Based on previous studies by the authors, the "Europe" experiment includes the updated parameterizations used over that region (Messmer et al., 2017), i.e., Noah-MP instead of Noah land surface scheme".*

*One could interpret that only the "Europe" run uses the Noah-MP scheme. But in previous lines it is stated that the Noah-MP LSM is used in all runs. Please clarify.*

As pointed out by the reviewer, the sentence was not clear enough. In the new version of the manuscript, we will change it to:

"*Based on previous studies by the authors, the "Europe" experiment includes the typical parameterizations used over that region (Messmer et al., 2017), but with the updated version of the Noah land surface scheme.*"

*L152-153:*

*"The "South America" experiment takes as a reference the parameterizations used to simulate storms over the central Andes (Zamuriano et al., 2019)."*

*The reference to Zamuriano et al. 2019 in https://nhess.copernicus.org/preprints/nhess-2019-286/ (search on December 20, 2021) appears as:*

*"Review status: this preprint was under review for the journal NHESS. A final paper is not foreseen. "*

*"This preprint has been withdrawn."*

*Even though the manuscript is available at: https://nhess.copernicus.org/preprints/nhess-2019-286/nhess-2019-286.pdf, the authors should re-consider (o better justify) the citation of this reference.*

Thank you for pointing this out. We will rephrase this sentence to justify this selection in a different way.

*Throughout the paper the authors use "the parameterization options" when referring to WRF simulations. It would be easier and more standard to read simply "the model" or "WRF", or "run".*

*Throughout the paper the authors write "monthly precipitation sums". It would be more standard to write "monthly accumulated precipitation".*

*For example, the sentence:*

*"The Europe parameterization option simulates especially low monthly precipitation sums in the wet seasons"*

*could be written:*

*"The Europe run simulates especially low monthly accumulated precipitation in the wet seasons"*

Thank you for these two suggestions, we will change these terminologies as suggested by the referee in the new version of the manuscript.

*L269-271:*

*"For 5-day and daily intervals the values drop for the correlations and rise for RMSEs. The increase in the RMSEs and the reduction in the correlations are expected due to the fact that capturing the exact amounts of precipitation at the same time as the observations is rather challenging for the model"*

*I would not say that this is challenging for the model, but a consequence of sensitivity to initial conditions (internal variability, present even if the model were perfect). These are not weather forecasts, but a continuous climate run.*

As we increase the temporal resolution of the analysis, the parameterization schemes play more and more an important role in the simulation, i.e., the exact point in time when a process such as cloud nucleation is invoked. As the simulation of this timing is a challenge for the regional climate model and the corresponding parameterizations, we argue that capturing the representation of the variables as observed with small temporal increments is more difficult for the model than capturing the coarser monthly values. From our perspective the internal variability should not play a major role, as the simulations are driven by the same reanalysis product, so the internal variability between the observations and the WRF runs should somehow align, as it is imposed through the boundary conditions of the model. We agree that this might play a role in case a real climate simulation is investigated, where the year 2008 in the model and in the observations would not be the same.

We will add parts of this explanation to the sentence pointed out by the referee to indicate which process is more challenging for the model to capture.

*In Figure 5 you say "The bold numbers indicate the best option for each region". The figure would be easier to read if the corresponding names of the experiments (on the left) were also in bold face.*

This is a valid suggestion and we will also change the names of the experiments to bold face with a larger number of months above the reference threshold.

*In addition, usually the symbols like the asterisks used in Figure 5 are used when a correlation is statistical significant.*

We agree with the referee that it is more common to use the asterisks to highlight the statistical significance of the results in the literature. However, as Fig. 5 is already complicated and full of details, in order to make it cleaner and clearer to the readers, we decided to use the asterisks to cross the statistically insignificant pattern correlations. The second column (the one related to the gridded datasets) is always significant, and otherwise, each pixel of that column would have been covered with the asterisks, and it would have complicated the interpretability of the figure.

*L336: change from "patters" to "patterns"*

Thank you for pointing out this typo, we will replace it as suggested.

*Is it possible to add Obs in Figure 8a?*

We will consider including the observations as well, but as we only have 15 stations compared to several thousands of grid points in the NE flatlands, we will check if and how such a comparison can be accomplished.

*It is not clear to me which domain it is being used for section 3.4 and for the previous sections. For sections 3.1 to 3.3 where the authors using results from D02? Are results of section 3.4 from the D03 domain? Please clarify.*

As already stated in one of the major comments we will be clearer about the fact that only results from D02 are shown in the analysis.

*L384. Which is the area for the computation of the field means? The entire D03 domain? Please clarify.*

Figure 8 was created by computing the field means of the different variables for the NE flatlands included in D02. This means that only the grid points with an elevation between 0 and 1000 metres were included in the analysis. We will clarify this in the new version of the manuscript and we will add figure R1.3 below to the supplementary material.

[Figure]

*Figure R1.3: Map of the second domain highlighting the five regions included in the analysis with different shading.*

*Figs 8 and 9. According to their captions, these figures refer to the NE slopes, but section 3.4 is devoted to flatlands. Please clarify.*

Thank you for pointing out this error. This is simply a typo in the titles of the panels. The lines correspond to the field means over the NE flatlands, and the title should state that. We will correct this error in the new version of the manuscript.

*In addition, Figure 8 is very interesting. Would it be possible to include a similar figure for some of the other(s) regions studied in Figure 3?*

As already stated in the corresponding major comment, we only analyzed the seasonal and daily cycles of the NE flatlands because our focus region is located in that area. We will consider including a new figure for the analysis over the plateau if the results are interesting and different enough, as the paper is already quite long.

*L396-399. The authors write*

*"The relative humidity and the precipitable water of the No Cumulus parameterization option is especially low, even though the precipitation is comparable to the other options, which might indicate that this parameterization option has an efficient process to remove moisture from the atmosphere,*

*i.e., convective processes. This is also supported by the fact that precipitation occurs mainly in the afternoon, while the other options have precipitation distributed over the whole day (Fig. 9)."*

*Do the results in Figs. 8 and 9 come from domains D01, D02 or D03?*

*In case results in Figs. 8 and 9 are from D02 or D03, please explain how the use of No-Cumulus in D01 is affecting the results in D02 or D03. In particular, one would expect that since D02 and D03 do not use a cumulus scheme in any of the simulations, the diurnal cycle would be the same, even in the No-Cumulus run. This is an interesting point that the authors could explain a bit better.*

As already stated before, we only analyzed the results from D02. We believe that the difference in the distribution of precipitation of the experiments is caused by the fact that only for NoCumulus the cumulus parameterization is switched off in D01, and not in the remaining runs. However, we will further investigate this by analyzing the daily cycle of the different variables over the same area (D02) in the first domain (D01), as the results of D01 are used as boundary conditions for D02. Hence, if moisture availability in D01 is shifted into the afternoon in the NoCumulus runs compared to in the morning, as in all the other runs, this will certainly affect the results in D02 and D03.

*L428-430. The authors write*

*"The region of interest is the entire department of Madre de Dios, but because of the lack of a dense network of weather stations in that area we evaluate the performance of the model over a broader area including the tri-national border of Peru, Bolivia and Brazil."*

*I find this comment rather unnecessary. The authors do a fine job at assessing the WRF simulations with the available data for both the broader region in D02 and for the smaller region in D03 (which they say is the region of interest). In this sense, both domains D02 and D03 are the region of interest according to the results and analyses presented in the paper. Maybe the authors could just write something like:*

*"The region of interest includes parts of the tri-national border of Peru, Bolivia and Brazil, with a focus on the region of Madre de Dios. The analysis of the latter is challenging given the lack of a dense network of weather stations in the area".*

Thank you for your suggestion, we will change it accordingly.

---

## Author Comment (AC2)

**Point-by-point Reply to Referee #2**

**Comment published online on the 4th of January, 2022**

*This work evaluates the performance of the Weather Research & Forecasting (WRF) model version 3.8.1 at convection-permitting scale over southern Peru using different configurations combining microphysics, cumulus, longwave radiations, and planetary boundary layer physic schemes. For this purpose, different comparisons for two years, 2008 and 2012, were performed between the WRF outputs and the observations of both weather stations and precipitation gridded products.*

*The topic addressed in this study is relevant, being the results here found of high value for the climate scientific community. The manuscript is well written and structured, with an appropriate discussion of the results, showing clear and concise conclusions. Therefore, I recommend this study for publication in the Geoscientific Model Development (GMD) journal after minor revisions. My comments are as follows:*

Thank you for taking your time to read our manuscript in detail and for your positive and constructive comments.

*Major comments:*

*1. The main analysis was based on 2008 as the precipitation over Madre de Dios was more or less standard (L437-438) in that year. In addition, the year 2012 was selected to test the model performance for wetter conditions. In this regard, as analysis focused on domain 2, how were 2008 and 2012 for the whole d02 domain? Were they also "standard" and "wet"? To clarify this point, it could be good to represent the climatology of domain 2 in Figure 2.*

We use PISCO as the main observational data set to perform that analysis. However, PISCO doesn't cover the entire domain 2 as stated in the manuscript. Thus, we decided to include the climatology for Madre de Dios in the paper instead of the one for the entire domain 2.

Figure R2.1 shows the analysis for the part of domain 2 included in PISCO (without the hatched area of Fig. 1). Even if a new domain is considered and a new precipitation mean is obtained for the same period of time (1981-2010), the annual precipitation anomalies (Fig. R2.1a) show that the year 2008 is a standard year in terms of precipitation, and that the year 2012 is also a wet year considering a larger domain. Additionally, Fig. R2.1b shows that the seasonal cycle is rather similar to the one presented in Fig. 2 in the manuscript, which means that well differentiated rainy and dry seasons are observable in both years compared to the 30-year climatology.

[Figure]

*Figure R2.1: (a) Annual precipitation anomalies (in millimetres per year), where the asterisks denotes our main study year 2008 and the triangle the wet year 2012, and (b) monthly accumulated values of precipitation (in millimetres per month) for years 2008 and 2012 (purple and orange horizontal lines, respectively) compared to the climatology (1981–2010, in grey, using a box and whisker plot).*

*2. For me, one of the most interesting parts of this study is the one related to section 3.4 (seasonal cycle over the northeastern flatland). For these analyses, a comparison could be made with observational values (for example, ERA5) to see which combination of parameterizations are closer to the "reality"? On the other hand, and just as a curiosity, why did the authors select IMERG and CHIRPS (and not the other two, i.e., ERA5 and TRMM) to compare with PISCO?*

This is something suggested also by referee 1. Hence, we replace the precipitation panels and add IMERG for the daily cycle (Fig. R2.2) and IMERG and observations for the seasonal cycle (Fig. R2.3). Fig R2.2 shows that Micro13, South America and Kenya are able to capture relatively well the precipitation of the first half of the day, but they all miss the peak in precipitation in the afternoon. Conversely, NoCumulus is able to capture the amount of precipitation during the afternoon correctly. Also the seasonal cycle shows a rather good alignment between IMERG and the observations and the

Micro13 run. We will replace these panels in the corresponding figures in the new version of the manuscript. We will not include ERA5, as it is not an observation-based data but a modeled reanalysis product, which is the driver of WRF and hence, it is not fully independent.

[Figure]

*Figure R2.2: Monthly mean daily cycle for July of a field mean over the northeastern flatlands for precipitation (mm) including also IMERG (pink line).*

[Figure]

*Figure R2.3: Seasonal cycle of a field mean over the northeastern flatlands for (a) monthly precipitation sums (mm/month) including also IMERG (pink line). The black line indicates the average of the station data and the gray shaded area indicates plus and minus one standard deviation.*

The reason why we generally focus on IMERG and CHIRPS is that these are the most advanced gridded observational data sets, including a variety of station data as well. TRMM is the predecessor of IMERG and hence not only the temporal but also the spatial resolution is better in IMERG, also the quality of the dataset itself must be assumed to be better in IMERG. ERA5 cannot be considered as a gridded observational dataset, as it is a reanalysis product, which is based on model simulations. Due

to the fact that ERA5 is the driving data set of the WRF runs, we prefer to compare the output of WRF to somewhat more independent data sets, i.e., CHIRPS and IMERG.

*3. Although the analyses proposed here at the monthly scale provide very valuable information on the characterization of the model and the different parameterization schemes, the use of the daily scale could provide additional information on how the model represents extreme values at a high spatial resolution, and which scheme is better in this regard for the different regions.*

As stated in the manuscript, the temporal analysis at finer temporal resolutions such as 15-days, 10-days, pentads or daily was also carried out for our analysis, and the same results are observed for the different intervals. In general, the RMSE increases and the temporal correlations decrease as we increase the temporal resolution. The statement in the manuscript is based on Fig. R2.4 below, that is neither shown in the manuscript nor in the supplementary material. Hence, we will add this figure for daily temporal resolution to the supplementary material.

[Figure]

*Figure R2.4: (a) The temporal correlation and (b) root-mean-square error (RMSE) between the annual cycle for the year 2008 of measured and simulated daily precipitation sums at the nearest grid point to the station's location shown for the different parameterization options and gridded observational datasets. The whiskers extend to the value that is no more than 1.5 times the inter-quartile range away from the box. The values outside this range are defined as outliers and are plotted with dots.*

For the evaluation of the representation of extreme values over each region, we plan to perform a short analysis on daily extreme precipitation values exceeding the 90th percentile.

*4. If I understood correctly, the results of domain 3 were not used for the evaluation of the model performance. I agree with the authors to focus the evaluation on domain d02, since, as mentioned by the authors in L428-430, there are not enough observations for a comparison at the finest spatial resolution (i.e., 1 km). However, due to the high computational cost required to run the model for an additional domain at 1-km, it might be good to better specify what was the final purpose of using a third domain, covering the department of Madre de Dios, in such a high spatial resolution.*

As stated in the manuscript, once the best configuration of the model has been determined, different climate simulations can be carried out to evaluate the effect of global warming or to investigate the interactions between the land and the atmosphere. In our case, we plan to investigate the effect of global warming over Madre de Dios, a region of Peru that can be considered a biodiversity hotspot and where the ecosystem provides everything that people need there (e.g., raw materials, fresh water, climate regulation, etc). At the same time, some threats are affecting the region such as illegal logging, deforestation or gold mining. These activities sustain to some extent the economy of the region, but at the same time they jeopardize the sustainable development of the region. New high resolution simulations over Madre de Dios will provide some insight about how the region is expected to change under climate conditions, and to infer the effect of those changes on the activities carried out in this biodiversity hotspot. With this in mind, it was also important to include the third domain in the tests, as some of the test runs include a two-way nesting configuration, which means that the result of the innermost domain influences the larger domains and vice versa.

We will include some new lines about this in the revised version manuscript to clarify the final purpose of the highly demanding third domain of the simulations.

***Minor comments:***

*- L121 "...so a spin-up period of two months is enough to balance the fluxes between the atmosphere and soil in WRF": Here, maybe, I would not affirm that a 2-month spin-up period is enough since, as the authors concluded, a longer spin-up period is probably needed for the simulations (L315-318 and L467-469).*

As we point out in the paper, we have performed a test with a longer spin-up period and we cannot see a systematic improvement of precipitation sums in the seasonal cycle, so a 2-month spin-up period should still be sufficient.

*- In section 2.1., please specify the number of vertical levels used in WRF, the top of the model, and the time-steps applied in simulation.*

All the sensitivity simulations include 49 vertical eta levels until the model top at 50 hPa, and the adaptive time step was employed while running the simulations. No nudging was applied to the input data. These details will be added to section 2.1 as suggested also by referee 1.

*- Please provide information about the time resolution of the weather station data in the main text.*

Thank you for pointing out this missing information. We will add it to the new manuscript:

"The weather station data from Peru are provided by the Servicio Nacional de Meteorología e Hidrología (SENAMHI) del Perú, the data from Bolivia by the SENAMHI Bolivia, and the data from Brazil by the Instituto Nacional de Meteorologia (INMET). These data are available with a daily temporal resolution."

*- L211: Did the authors check other interpolation methods? Please justify why bilinear interpolation was used instead of others (e.g., nearest neighbor).*

We have not considered other interpolation methods. The bilinear interpolation is widely used in the literature, and that is the method that we usually follow. We think that the results are insensitive to the interpolation method selected, but we will redo some of the analysis performing also the nearest-neighbour interpolation to assess the sensitivity of the results to the chosen interpolation method.

*- L327 "In the NE flatlands, the pattern correlation is rather good compared to the temporal correlation": Here, it could be good to remember that the comparison is between the results from Figure 5 and Figure 3.*

We will add the reference to these two figures in the new version of the manuscript, as suggested by the referee.

*- L346-347 "However, No Cumulus shows a general excess of precipitation in the whole domain" and L378 "... except for the last which overestimates the amount": It is hard for me to see that the No Cumulus combination generates more precipitation, in general, for the entire domain than others parameterization schemes (e.g., Micro13). In this regard, it could be good to indicate the mean value of the accumulated monthly precipitation for the entire domain, or even the mean for the different five regions.*

We will perform this analysis for each of the runs and different regions, and provide the numbers in form of a table, which will either be shown in the manuscript or in the supplementary material.

*- L356-359 "Except for the Micro13 parameterization option, most of the simulations...in the NE flatlands compared to the other parameterization options": Were the authors referring to the results obtained from the transect? Please, clarify this point.*

Yes, we refer to the transect as stated in line 350. We will clarify that in the new version of the manuscript.

*- L363-364 "For the plateau, the simulations agree with PISCO on the rather dry conditions, except for Micro13 and CHIRPS that show wetter patterns": Here, I would suggest removing the information for*

*CHIRPS as the comparison seems to be between simulations and PISCO. Otherwise, I would change the sentence to express it in another way.*

As noted by the referee, the comparison is between simulations and PISCO. We will reformulate this sentence in the new manuscript to:

"For the plateau, the simulations agree with PISCO on the rather dry conditions, except for Micro13 that shows wetter patterns. These wetter conditions are also represented by CHIRPS."

***Figures***

*- Figure 3a-e: I would suggest changing the color of the box for CHIRPS. Here, the median is sometimes hard to see.*

We agree that the median is hard to see in the boxes for CHIRPS. However, as we had a hard time to select well distinguishable colors (all the gridded observational data sets, except for PISCO, share the same color family), we will color the median in white and leave the color as it is.

*- Figure 3f-l: I would suggest changing the colors of the lines bordering the different regions. It is sometimes difficult to differentiate between the borders of the regions (i.e., plateau, SW slopes, SW flatlands, NE slopes, and NE flatlands) and the borders of Madre de Dios. Also, if not necessary, it could be good to remove the black lines in all the maps. Do these lines represent the borders between countries?*

As identified by the referee, the black lines in the maps represent the country borders. We will consider removing those lines from the plots, and include only the border of Madre de Dios together with the lines bordering the five regions. The last will then be colored in grey.

*- Figure 4b: I would suggest changing the range on the y-axis for this case to better show the box and whisker plot.*

In Fig. 4, all the box and whisker plots share the same range in the y-axis. This facilitated finding the regions with the largest and lowest RMSEs. We have tested this for panel 4d (we think the referee refers to Fig. 4d instead of 4b), but as it does not give more information to the reader, we would like to keep it as it is in the current version of the manuscript and to have the same range for all the y-axes.

[Figure]

*Figure R2.5: As Fig. 4d in the manuscript, but with a smaller y-axis range.*

*- Figures 6 and 7: I would suggest adding the borders between the five regions (as in Figures 3 and 4) in order to better follow the discussion of the results.*

As for Fig. 3 and 4 we will remove the country borders and add instead the lines bordering the five different regions.

---

## Author Response (AR1)

**Point-by-point Reply to Referee #1**

**Comment published online on the 20th of December, 2021**

*In their work, González-Rojí et al. present an assessment of the WRF model skill at simulating precipitation over parts of South America, with a focus on parts of Peru, Brazil and Bolivia. The authors explore different configurations of the WRF model, evaluating the performance of the model by comparing precipitation outputs to observations from weather stations and estimates from observation-based products and ERA5. From their analysis, the authors are able to identify which of the studied model configurations work best for their region of interest. In addition, the authors identify some strengths and weaknesses from observation-based products, like PISCO and CHIRPS.*

*The manuscript is well written and the topic is very relevant both for the climate science community of South America and the convection-permitting modeling community. I recommend this study for publication after minor revisions. Please see the details below.*

Thank you for reading our manuscript so carefully and for your positive and constructive comments.

*Major comments:*

*1. Taking further advantage of the high resolution simulations:*

*Simulations at such high resolution are very valuable for the region of interest. The analysis of the monthly accumulated precipitation is very interesting. A further analysis of 5-day or daily accumulated precipitation would help to strengthen the paper. For example, the analysis of the statistical distributions of the daily accumulated precipitation would help to identify the value of high resolution simulations at representing extreme precipitation.*

As stated in the manuscript, the temporal analysis at finer temporal resolutions such as 15-days, 10-days, pentads or daily was also carried out for our analysis, and the same results are observed for the different intervals. In general, the RMSE increases and the temporal correlations decrease as we increase the temporal resolution. The statement in the manuscript is based on Fig. R1.1, that is neither shown in the manuscript nor in the supplementary material. Hence, we have added this figure to the supplementary material.

[Figure]

*Figure R1.1: (a) The temporal correlation and (b) root-mean-square error (RMSE) between the annual cycle for the year 2008 of measured and simulated daily precipitation sums at the nearest grid point to the station's location shown for the different parameterization options and gridded observational datasets. The whiskers extend to the value that is no more than 1.5 times the inter-quartile range away from the box. The values outside this range are defined as outliers and are plotted with dots.*

*In addition, in the last section about the diurnal cycle, it would be interesting to add the observations, if hourly data is available.*

We agree that it would be interesting to include the observations to the plot, but the weather station data obtained for the validation of the sensitivity experiments is only available with daily temporal resolution. For precipitation we have added the data of IMERG (Fig. R1.2), as it is available with 30 min temporal resolution, but it must be noted that IMERG is not the best gridded data set available in any of the five regions studied. For all the other variables we are not aware of any data set with higher than daily temporal resolution. Figure R1.2 shows that Micro13, South America and Kenya are able to capture relatively well the precipitation of the first half of the day, but they all miss the peak in precipitation in the afternoon. Conversely, NoCumulus is able to simulate a peak of precipitation during the afternoon. We have replaced this panel in Fig. 9 of the manuscript.

[Figure]

*Figure R1.2: Monthly mean daily cycle for July of a field mean over the northeastern flatlands for precipitation (mm) including also IMERG (pink line).*

*2. More details about the configuration and domains:*

*Include some other standard details about the simulations, like the number of vertical levels, model top, type of nudging (if used). In addition, please be more explicit about which of the domains is used in each part of the paper (for example state explicitly whether results from D02 or D03 are used in section 3.4, and so forth).*

All the sensitivity simulations include 49 vertical eta levels until the model top at 50 hPa, and the adaptive time step was employed while running the simulations. No nudging was applied to the input data. These details were added to section 2.1 as suggested. Additionally, we added at the beginning of each section of the results that we were only considering the second domain (D02) of the simulations in the analysis.

*3. A small extension of the relationship between variables:*

*Both figures 8 and 9 are very interesting, as they allow to talk about possible relations between variables looking for an explanation about the behaviour of the simulated precipitation. It would be very interesting to see this analysis extended to at least one of the other regions studied in Figure 3. The authors could select for comparison, for example, the regions in flatlands vs. regions over the plateau, or over the slopes, where the simulated cloud field (both in terms of magnitude of cloud fraction and periodicities -e.g. annual and diurnal cycle-) might be qualitatively different.*

Figure R1.3 shows the seasonal cycle of a field mean over the plateau for precipitation, 2-metre temperature, precipitable water, mid-level cloud fraction, 2-metre relative humidity and soil moisture of the first layer of the land model. Figure R1.4 shows the daily cycle for July of a field mean over the plateau for precipitation and temperature. Compared to the seasonal cycle over the northeastern flatlands, small differences can be observed at the beginning of the year for the plateau. As explained in the manuscript, these are caused by the feedback between mid level clouds, soil moisture, precipitable water and precipitation.

We think that these figures should neither be included in the new version of the manuscript nor in the supplementary material, since they do not show any new results that improve the outcome of our paper.

[Figure]

*Figure R1.3: Seasonal cycle of a field mean over the plateau for (a) monthly accumulated precipitation (mm), (b) monthly mean of 2-metre temperature (K), (c) monthly mean of precipitable water (kg m⁻², (d) monthly mean fraction of mid-level (642-350 hPa) clouds (%), (e) monthly mean of 2-metre relative humidity (%), and (f) monthly mean soil moisture in the first soil layer (0-10 cm, m³m⁻³). The monthly accumulated precipitation (a) includes IMERG as a pink line. Additionally, the black line indicates the average of the weather station data and the gray shaded area indicates plus and minus one standard deviation. The x-axis shows the months of the year.*

[Figure]

*Figure R1.4: Daily cycle for July of a field mean over the plateau for (a) precipitation (mm), and (b) 2-metre temperature (K). The daily cycle of precipitation (a) also includes IMERG, depicted with a pink line. The x-axis shows the hours of the day, shifted by 5 hours from UTC to show local Peruvian time.*

*4. A comment about the order of the sections:*

*The manuscript is well written, and the sections are clear. However, it seems to me more clear to start with section 3.3, where the mean biases are presented, and then go to sections 3.1 (temporal correlations) and 3.2 (spatial correlations) where second-order metrics (correlations and RMSEs) are studied.*

We have evaluated whether changing the structure of the paper would increase the readability of the paper, and we have decided to keep it the way we submitted it because otherwise we change the storyline of the manuscript. The analysis of the RMSEs and the temporal correlations of the different experiments and gridded observational data sets against weather station data allows to select those gridded data sets with the best performance. The selected gridded data sets are used as reference in the next steps and provide the basis for the analysis of the pattern correlations against gridded data sets. If we change the structure, this would not be so clear for the reader, and consequently, we have decided not to change the structure.

*Minor comments:*

*L147-149. The authors write:*

*"Based on previous studies by the authors, the "Europe" experiment includes the updated parameterizations used over that region (Messmer et al., 2017), i.e., Noah-MP instead of Noah land surface scheme".*

*One could interpret that only the "Europe" run uses the Noah-MP scheme. But in previous lines it is stated that the Noah-MP LSM is used in all runs. Please clarify.*

As pointed out by the reviewer, the sentence was not clear enough. In the new version of the manuscript, we have changed it to:

"*Based on previous studies by the authors, the "Europe" experiment includes the typical parameterizations used over that region (Messmer et al., 2017), but with the updated version of the Noah land surface scheme.*"

*L152-153:*

*"The "South America" experiment takes as a reference the parameterizations used to simulate storms over the central Andes (Zamuriano et al., 2019)."*

*The reference to Zamuriano et al. 2019 in https://nhess.copernicus.org/preprints/nhess-2019-286/ (search on December 20, 2021) appears as:*

*"Review status: this preprint was under review for the journal NHESS. A final paper is not foreseen. "*

*"This preprint has been withdrawn."*

*Even though the manuscript is available at: https://nhess.copernicus.org/preprints/nhess-2019-286/nhess-2019-286.pdf, the authors should re-consider (o better justify) the citation of this reference.*

Thank you for pointing this out. When we started writing the first draft of this paper, this manuscript was still under review, so we could not know that it was not foreseen for publication. We have contacted one of the authors of the manuscript, and got reassured that the withdrawal is not related to scientific reasons. The referee comments about the manuscript are generally positive as anyone can see. We think that the preprint is still citable, as it shows results about a solid WRF simulation over the Andes, and thus we decided to keep the citation in the manuscript.

*Throughout the paper the authors use "the parameterization options" when referring to WRF simulations. It would be easier and more standard to read simply "the model" or "WRF", or "run".*

*Throughout the paper the authors write "monthly precipitation sums". It would be more standard to write "monthly accumulated precipitation".*

*For example, the sentence:*

*"The Europe parameterization option simulates especially low monthly precipitation sums in the wet seasons"*

*could be written:*

*"The Europe run simulates especially low monthly accumulated precipitation in the wet seasons"*

Thank you for these two suggestions, we have changed these terminologies as suggested by the referee in the new version of the manuscript.

*L269-271:*

*"For 5-day and daily intervals the values drop for the correlations and rise for RMSEs. The increase in the RMSEs and the reduction in the correlations are expected due to the fact that capturing the exact amounts of precipitation at the same time as the observations is rather challenging for the model"*

*I would not say that this is challenging for the model, but a consequence of sensitivity to initial conditions (internal variability, present even if the model were perfect). These are not weather forecasts, but a continuous climate run.*

As we increase the temporal resolution of the analysis, the parameterization schemes play more and more important role in the simulation, i.e., the exact point in time when a process such as cloud nucleation is invoked. As the simulation of this timing is a challenge for the regional climate model and the corresponding parameterizations, we argue that capturing the representation of the variables as observed with small temporal increments is more difficult for the model than capturing the coarser monthly values. From our perspective the internal variability should not play a major role, as the simulations are driven by a reanalysis product, so the internal variability between the observations and the WRF runs should somehow align, as the reanalysis is imposed through the boundary conditions of the model. We agree that this might play a role in case a real climate simulation is investigated, where the year 2008 in the model and in the observations would not be the same.

To clarify this point in the new version of the manuscript, we have added some parts of this explanation to the sentence pointed out by the referee.

*In Figure 5 you say "The bold numbers indicate the best option for each region". The figure would be easier to read if the corresponding names of the experiments (on the left) were also in bold face.*

This is a valid suggestion and we have changed the names of the experiments to bold face with a larger number of months above the reference threshold.

*In addition, usually the symbols like the asterisks used in Figure 5 are used when a correlation is statistical significant.*

We agree with the referee that it is more common to use the asterisks to highlight the statistical significance of the results in the literature. However, as Fig. 5 is already complicated and full of details, in order to make it cleaner and clearer to the readers, we decided to use the asterisks to indicate the statistically insignificant pattern correlations. The second column (the one related to the gridded datasets) is always significant; if each pixel of that column is covered by an asterisk, it would complicate the interpretability of the figure. That is why we have decided to keep the asterisks as they were in the previous version of the manuscript.

*L336: change from "patters" to "patterns"*

Thank you for pointing out this typo, we have corrected it in the new version of the manuscript.

*Is it possible to add Obs in Figure 8a?*

We have included the observations as well to Figure 8a. In the new figure, we show the mean of the weather station data and the corresponding plus/minus one standard deviation.

*It is not clear to me which domain it is being used for section 3.4 and for the previous sections. For sections 3.1 to 3.3 where the authors using results from D02? Are results of section 3.4 from the D03 domain? Please clarify.*

As already stated in one of the major comments, we have added at the beginning of each subsection of the results that we only show results from D02 in the analysis.

*L384. Which is the area for the computation of the field means? The entire D03 domain? Please clarify.*

Figure 8 was created by computing the field means of the different variables for the NE flatlands included in D02. This means that only the grid points with an elevation between 0 and 1000 metres were included in the analysis. We have clarified this in the new version of the manuscript and we have added figure R1.5 below to the supplementary material.

[Figure]

*Figure R1.5: Map of the second domain highlighting the five regions included in the analysis with different shading.*

*Figs 8 and 9. According to their captions, these figures refer to the NE slopes, but section 3.4 is devoted to flatlands. Please clarify.*

Thank you for pointing out this error. This is simply a typo in the titles of the panels. The lines correspond to the field means over the NE flatlands, and the title should state that. We have corrected this error in the new version of the manuscript.

*In addition, Figure 8 is very interesting. Would it be possible to include a similar figure for some of the other(s) regions studied in Figure 3?*

As already stated in the corresponding major comment, we have analyzed the seasonal and daily cycles over the plateau (see Fig. R1.3 and R1.4). However, we have decided not to include any of those

figures in the new version of the manuscript nor in the supplementary information, since they do not show any remarkable result significant for the outcome of the study.

*L396-399. The authors write*

*"The relative humidity and the precipitable water of the No Cumulus parameterization option is especially low, even though the precipitation is comparable to the other options, which might indicate that this parameterization option has an efficient process to remove moisture from the atmosphere, i.e., convective processes. This is also supported by the fact that precipitation occurs mainly in the afternoon, while the other options have precipitation distributed over the whole day (Fig. 9)."*

*Do the results in Figs. 8 and 9 come from domains D01, D02 or D03?*

*In case results in Figs. 8 and 9 are from D02 or D03, please explain how the use of No-Cumulus in D01 is affecting the results in D02 or D03. In particular, one would expect that since D02 and D03 do not use a cumulus scheme in any of the simulations, the diurnal cycle would be the same, even in the No-Cumulus run. This is an interesting point that the authors could explain a bit better.*

As already stated before, we only analyzed the results from D02. However, we have further investigated this by analyzing the daily cycle of the different variables over the same area (D02) in the first domain (D01), as the results of D01 are used as boundary conditions for D02 (Figure R1.6). Since two-way nesting was used for the South America and Europe runs, they show the same daily cycle because the results from the nested domains are overwritten onto the respective parent domain. In both domains, the Kenya and Micro13 runs show similar daily cycles, with the same difference in precipitation amounts over the entire day. The cycle of NoCumulus is different in D01 to that in D02, and in D01 the maximum is also shifted to the afternoon compared to the other runs. Thus, the difference in the distribution of precipitation of the experiments is caused by the fact that only for NoCumulus the cumulus parameterization is switched off in D01, and not in the remaining runs.

We believe that the figure is not important enough to be included in the manuscript, but it informs about the behaviour of not using cumulus parameterizations schemes in any of the domains. We have included a line about this result in the corresponding part of the results.

[Figure]

Figure R1.6: Daily cycle for July of a field mean over the northeastern flatlands of D02 for (a) precipitation (mm), and (b) 2-metre temperature (K). (c ) and (d) show the same variables respectively, but over the same area (D02) in the first domain (D01).  The daily cycles of precipitation (a and c) also include IMERG, depicted with a pink line. The x-axis shows the hours of the day, shifted by 5 hours from UTC to show local Peru time.

*L428-430. The authors write*

*"The region of interest is the entire department of Madre de Dios, but because of the lack of a dense network of weather stations in that area we evaluate the performance of the model over a broader area including the tri-national border of Peru, Bolivia and Brazil."*

*I find this comment rather unnecessary. The authors do a fine job at assessing the WRF simulations with the available data for both the broader region in D02 and for the smaller region in D03 (which they say is the region of interest).  In this sense, both domains D02 and D03 are the region of interest according to the results and analyses presented in the paper.  Maybe the authors could just write something like:*

*"The region of interest includes parts of the tri-national border of Peru, Bolivia and Brazil, with a focus on the region of Madre de Dios. The analysis of the latter is challenging given the lack of a dense network of weather stations in the area".*

Thank you for your suggestion, we have changed it accordingly.

**Point-by-point Reply to Referee #2**

**Comment published online on the 4th of January, 2022**

*This work evaluates the performance of the Weather Research & Forecasting (WRF) model version 3.8.1 at convection-permitting scale over southern Peru using different configurations combining microphysics, cumulus, longwave radiations, and planetary boundary layer physic schemes. For this purpose, different comparisons for two years, 2008 and 2012, were performed between the WRF outputs and the observations of both weather stations and precipitation gridded products.*

*The topic addressed in this study is relevant, being the results here found of high value for the climate scientific community. The manuscript is well written and structured, with an appropriate discussion of the results, showing clear and concise conclusions. Therefore, I recommend this study for publication in the Geoscientific Model Development (GMD) journal after minor revisions. My comments are as follows:*

Thank you for taking your time to read our manuscript in detail and for your positive and constructive comments.

*Major comments:*

*1. The main analysis was based on 2008 as the precipitation over Madre de Dios was more or less standard (L437-438) in that year. In addition, the year 2012 was selected to test the model performance for wetter conditions. In this regard, as analysis focused on domain 2, how were 2008 and 2012 for the whole d02 domain? Were they also "standard" and "wet"? To clarify this point, it could be good to represent the climatology of domain 2 in Figure 2.*

We use PISCO as the main observational data set to perform that analysis. However, PISCO doesn't cover the entire domain 2 as stated in the manuscript. Thus, we decided to include the climatology for Madre de Dios in the paper instead of the one for the entire domain 2.

Figure R2.1 shows the analysis for the part of domain 2 included in PISCO (without the hatched area of Fig. 1). Even if a new domain is considered and a new precipitation mean is obtained for the same period of time (1981-2010), the annual precipitation anomalies (Fig. R2.1a) show that the year 2008 is a standard year in terms of precipitation, and that the year 2012 is also a wet year considering a larger domain. Additionally, Fig. R2.1b shows that the seasonal cycle is rather similar to the one presented in Fig. 2 in the manuscript, which means that well differentiated rainy and dry seasons are observable in both years compared to the 30-year climatology.

[Figure]

*Figure R2.1: (a) Annual precipitation anomalies (in millimetres per year), where the asterisks denotes our main study year 2008 and the triangle the wet year 2012, and (b) monthly accumulated values of precipitation (in millimetres per month) for years 2008 and 2012 (purple and orange horizontal lines, respectively) compared to the climatology (1981–2010, in grey, using a box and whisker plot).*

*2. For me, one of the most interesting parts of this study is the one related to section 3.4 (seasonal cycle over the northeastern flatland). For these analyses, a comparison could be made with observational values (for example, ERA5) to see which combination of parameterizations are closer to the "reality"? On the other hand, and just as a curiosity, why did the authors select IMERG and CHIRPS (and not the other two, i.e., ERA5 and TRMM) to compare with PISCO?*

This is something suggested also by referee 1. Hence, we replaced the precipitation panels and added IMERG for the daily cycle (Fig. R2.2), and IMERG and observations for the seasonal cycle (Fig. R2.3). Fig R2.2 shows that Micro13, South America and Kenya are able to capture relatively well the precipitation of the first half of the day, but they all miss the peak in precipitation in the afternoon. Conversely, NoCumulus is able to capture the amount of precipitation during the afternoon correctly. Also the seasonal cycle shows a rather good alignment between IMERG and the observations and the

Micro13 run. We have replaced these panels in the corresponding figures in the new version of the manuscript. However, we have not included ERA5, as it is not an observation-based data but a modeled reanalysis product, which is the driver of WRF and hence, it is not fully independent.

[Figure]

*Figure R2.2: Monthly mean daily cycle for July of a field mean over the northeastern flatlands for precipitation (mm) including also IMERG (pink line).*

[Figure]

*Figure R2.3: Seasonal cycle of a field mean over the northeastern flatlands for (a) monthly precipitation sums (mm/month) including also IMERG (pink line). The black line indicates the average of the station data and the gray shaded area indicates plus and minus one standard deviation.*

The reason why we generally focus on IMERG and CHIRPS is that these are the most advanced gridded observational data sets, including a variety of station data as well. TRMM is the predecessor of IMERG and hence not only the temporal but also the spatial resolution is better in IMERG, also the quality of the dataset itself must be assumed to be better in IMERG. ERA5 cannot be considered as a gridded observational dataset, as it is a reanalysis product, which is based on model simulations. Due to the

fact that ERA5 is the driving data set of the WRF runs, we prefer to compare the output of WRF to somewhat more independent data sets, i.e., CHIRPS and IMERG.

*3. Although the analyses proposed here at the monthly scale provide very valuable information on the characterization of the model and the different parameterization schemes, the use of the daily scale could provide additional information on how the model represents extreme values at a high spatial resolution, and which scheme is better in this regard for the different regions.*

As stated in the manuscript, the temporal analysis at finer temporal resolutions such as 15-days, 10-days, pentads or daily was also carried out for our analysis, and the same results are observed for the different intervals. In general, the RMSE increases and the temporal correlations decrease as we increase the temporal resolution. The statement in the manuscript is based on Fig. R2.4 below, that is neither shown in the manuscript nor in the supplementary material. Hence, we have added this figure for daily temporal resolution to the supplementary material.

[Figure]

*Figure R2.4: (a) The temporal correlation and (b) root-mean-square error (RMSE) between the annual cycle for the year 2008 of measured and simulated daily precipitation sums at the nearest grid point to the station's location shown for the different parameterization options and gridded observational datasets. The whiskers extend to the value that is no more than 1.5 times the inter-quartile range away from the box. The values outside this range are defined as outliers and are plotted with dots.*

For the analysis on daily extreme precipitation, we have calculated the 90th, 95th and the 99th percentile of each region during every month. This short analysis shows that the largest 90th, 95th and 99th percentiles in every month are found in the NE slopes, followed by the NE flatlands and the plateau. Among the runs, Micro13 shows the largest values and Europe the lowest. Since this is something that can be expected from the previous results of the analysis, we have decided not to include this in the new version of the manuscript.

*4. If I understood correctly, the results of domain 3 were not used for the evaluation of the model performance. I agree with the authors to focus the evaluation on domain d02, since, as mentioned by the authors in L428-430, there are not enough observations for a comparison at the finest spatial resolution (i.e., 1 km). However, due to the high computational cost required to run the model for an additional domain at 1-km, it might be good to better specify what was the final purpose of using a third domain, covering the department of Madre de Dios, in such a high spatial resolution.*

As stated in the manuscript, once the best configuration of the model has been determined, different climate simulations can be carried out to evaluate the effect of global warming or to investigate the interactions between the land and the atmosphere. In our case, we plan to investigate the effect of global warming over Madre de Dios, a region of Peru that can be considered a biodiversity hotspot and where the ecosystem provides everything that people need there (e.g., raw materials, fresh water, climate regulation, etc). At the same time, some threats affect the region such as illegal logging, deforestation or gold mining. These activities sustain to some extent the economy of the region, but at the same time they jeopardize the sustainable development of the region. New high resolution simulations over Madre de Dios will provide some insight about how the region is expected to change under climate conditions, and to infer the effect of those changes on the activities carried out in this biodiversity hotspot. With this in mind, it is also important to include the third domain in the tests, as some of the test runs include a two-way nesting configuration, which means that the result of the innermost domain influences the larger domains and vice versa.

We have included some new lines about this in the revised manuscript to clarify the final purpose of the highly demanding third domain of the simulations, particularly in the WRF model section and the conclusions.

*Minor comments:*

*- L121 "...so a spin-up period of two months is enough to balance the fluxes between the atmosphere and soil in WRF": Here, maybe, I would not affirm that a 2-month spin-up period is enough since, as the authors concluded, a longer spin-up period is probably needed for the simulations (L315-318 and L467-469).*

As we point out in the paper, we have performed a test with a longer spin-up period and we cannot see a systematic improvement of precipitation sums in the seasonal cycle, so a 2-month spin-up period should still be sufficient.

*- In section 2.1., please specify the number of vertical levels used in WRF, the top of the model, and the time-steps applied in simulation.*

All the sensitivity simulations include 49 vertical eta levels until the model top at 50 hPa, and the adaptive time step was employed while running the simulations. No nudging was applied to the input data. These details were added to section 2.1 in the new version of the manuscript.

*- Please provide information about the time resolution of the weather station data in the main text.*

Thank you for pointing out this missing information. We have added it to the new manuscript:

"The weather station data from Peru are provided by the Servicio Nacional de Meteorología e Hidrología (SENAMHI) del Perú, the data from Bolivia by the SENAMHI Bolivia, and the data from Brazil by the Instituto Nacional de Meteorologia (INMET). These data are available with a daily temporal resolution."

*- L211: Did the authors check other interpolation methods? Please justify why bilinear interpolation was used instead of others (e.g., nearest neighbor).*

We have redone the temporal analysis included in our study, but applying the nearest neighbor interpolation this time. Fig. R2.5 and R2.6 show the temporal Spearman correlations and RMSEs over the entire second domain and each different region obtained for the tested WRF runs and the gridded observational data sets. In each figure, the first line (a) is produced after applying the bilinear interpolation to the gridded observational data sets, and the second line (b) is produced after applying the nearest neighbor technique. Even though some distributions look a bit different for the two interpolation methods, e.g., PISCO has a narrower distribution when using the nearest neighbor method in the SW flatlands, the overall result is not affected by the selected interpolation method. Hence, also with the nearest neighbor interpolation technique, the same results as presented in the manuscript arise from the analysis: CHIRPS is highlighted as the best performing observational data set over the NE flatlands and slopes, while PISCO is preferred over the plateau and the SW slopes and flatlands. We believe that these figures should not be included in the manuscript nor the supplementary materials, but we have included a line in the manuscript highlighting that these results are insensitive to the interpolation technique applied.

[Figure]

**(a)** Correlations – Monthly data

[Figure]

**(b)** Correlations – Monthly data – Nearest Neighbor

Europe    Kenya    Micro13    CHIRPS    TRMM
South America    No Cumulus    ERA5    IMERG    PISCO

*Figure R2.5: The temporal correlation between the annual cycle for the year 2008 of measured and simulated daily precipitation sums at the nearest grid point to the station's location shown for the different parameterization options and gridded observational datasets after applying on them (a) the bilinear interpolation and (b) the nearest neighbor techniques. The whiskers extend to the value that is no more than 1.5 times the inter-quartile range away from the box. The values outside this range are defined as outliers and are plotted with dots.*

**(a)**

[Figure]

*Figure R2.6: Same as Figure R2.5, but for the root-mean-square error (RMSE).*

*- L327 "In the NE flatlands, the pattern correlation is rather good compared to the temporal correlation": Here, it could be good to remember that the comparison is between the results from Figure 5 and Figure 3.*

We have added the reference to these two figures in the new version of the manuscript, as suggested by the referee.

*- L346-347 "However, No Cumulus shows a general excess of precipitation in the whole domain" and L378 "... except for the last which overestimates the amount": It is hard for me to see that the No Cumulus combination generates more precipitation, in general, for the entire domain than others parameterization schemes (e.g., Micro13). In this regard, it could be good to indicate the mean value of the accumulated monthly precipitation for the entire domain, or even the mean for the different five regions.*

We have calculated the monthly mean precipitation for each run over the different areas included in the second domain of the setup. Table R2.1 shows the values for February. We have modified the sentences highlighted by the referee to be more specific about the regions where the excess or lack of precipitation is observed in each run. We have included this table, and the one for July 2008 in the supplementary materials.

*Table R2.1: Mean accumulated precipitation for February 2008 of each run over the different regions of the second domain: NE flatlands and slopes, the plateau, and the SW flatlands and slopes.*

| Experiment | NE flatlands | NE slopes | Plateau | SW slopes | SW flatlands |
|---|---|---|---|---|---|
| Europe | 101.2 | 276.03 | 116.22 | 4.59 | 1.18 |
| South America | 241.22 | 477.15 | 170.67 | 6.59 | 0.63 |
| Kenya | 167.76 | 482.18 | 168.61 | 6.87 | 2.24 |
| No Cumulus | 205.66 | 395.34 | 219.56 | 10.74 | 2.28 |
| Micro13 | 269.71 | 595.73 | 270.82 | 50 | 2.08 |
| CHIRPS | 265.44 | 244.33 | 119.05 | 48.05 | 26.94 |
| PISCO | 259.18 | 353.58 | 126.95 | 16.28 | 2.13 |

*- L356-359 "Except for the Micro13 parameterization option, most of the simulations...in the NE flatlands compared to the other parameterization options": Were the authors referring to the results obtained from the transect? Please, clarify this point.*

Yes, we refer to the transect as stated in line 350. We made that clear in the new version of the manuscript.

*- L363-364 "For the plateau, the simulations agree with PISCO on the rather dry conditions, except for Micro13 and CHIRPS that show wetter patterns": Here, I would suggest removing the information for CHIRPS as the comparison seems to be between simulations and PISCO. Otherwise, I would change the sentence to express it in another way.*

As noted by the referee, the comparison is between simulations and PISCO. We have reformulated this sentence in the new manuscript to:

"For the plateau, the runs agree with PISCO on the rather dry conditions, except for Micro13 that shows wetter patterns. These wetter conditions are also represented by CHIRPS."

**Figures**

*- Figure 3a-e: I would suggest changing the color of the box for CHIRPS. Here, the median is sometimes hard to see.*

We agree that the median is hard to see in the boxes for CHIRPS. However, as we had a hard time to select well distinguishable colors (all the gridded observational data sets, except for PISCO, share the same color family), we haven't changed the color of CHIRPS, and conversely, we have colored the median in white in both Figs. 3 and 4 .

*- Figure 3f-l: I would suggest changing the colors of the lines bordering the different regions. It is sometimes difficult to differentiate between the borders of the regions (i.e., plateau, SW slopes, SW flatlands, NE slopes, and NE flatlands) and the borders of Madre de Dios. Also, if not necessary, it could be good to remove the black lines in all the maps. Do these lines represent the borders between countries?*

As identified by the referee, the black lines in the maps represent the country borders. We have removed those lines from the plots, and include only the border of Madre de Dios together with the lines bordering the five regions. The last are now colored in grey.

*- Figure 4b: I would suggest changing the range on the y-axis for this case to better show the box and whisker plot.*

In Fig. 4, all the box and whisker plots share the same range in the y-axis. This facilitates finding the regions with the largest and lowest RMSEs. We have tested this for panel 4d (we think the referee refers to Fig. 4d instead of 4b) in Fig. R2.7, but as it does not give more information to the reader, we have kept it as it was in the new version of the manuscript and to have the same range for all the y-axes.

[Figure]

*Figure R2.7: As Fig. 4d in the manuscript, but with a smaller y-axis range.*

*- Figures 6 and 7: I would suggest adding the borders between the five regions (as in Figures 3 and 4) in order to better follow the discussion of the results.*

As for Fig. 3 and 4 we have removed the country borders and added instead the lines bordering the five different regions.